# The DNA binding landscape of the maize AUXIN RESPONSE FACTOR family

Mary Galli[1], Arjun Khakhar[2], Zefu Lu[3], Zongliang Chen[1], Sidharth Sen [4], Trupti Joshi[4,5], Jennifer L. Nemhauser[2], Robert J. Schmitz[3] & Andrea Gallavotti [1,6]

AUXIN RESPONSE FACTORS (ARFs) are plant-specific transcription factors (TFs) that couple perception of the hormone auxin to gene expression programs essential to all land plants. As with many large TF families, a key question is whether individual members determine developmental specificity by binding distinct target genes. We use DAP-seq to generate genome-wide in vitro TF:DNA interaction maps for fourteen maize ARFs from the evolutionarily conserved A and B clades. Comparative analysis reveal a high degree of binding site overlap for ARFs of the same clade, but largely distinct clade A and B binding. Many sites are however co-occupied by ARFs from both clades, suggesting transcriptional coordination for many genes. Among these, we investigate known QTLs and use machine learning to predict the impact of *cis*-regulatory variation. Overall, large-scale comparative analysis of ARF binding suggests that auxin response specificity may be determined by factors other than individual ARF binding site selection.

[1] Waksman Institute of Microbiology, Rutgers University, Piscataway, NJ 08854-8020, USA. [2] Department of Biology, University of Washington, Seattle, WA 98195-1800, USA. [3] Department of Genetics, The University of Georgia, Athens, GA 30602, USA. [4] Informatics Institute, University of Missouri, Columbia, MO 65211, USA. [5] Department of Health Management and Informatics and Christopher S. Bond Life Science Center, University of Missouri, Columbia, MO 65211, USA. [6] Department of Plant Biology, Rutgers University, New Brunswick, NJ 08901, USA. Correspondence and requests for materials should be addressed to A.G. (email: agallavotti@waksman.rutgers.edu)

Eukaryotic nuclear hormone signaling induces specific transcriptional changes crucial for many biological processes. While plant nuclear hormone signaling proteins differ substantially from those of animals, their pathways share many unexplained features including the ability to elicit diverse developmental responses in different cell types. In no plant hormone signaling pathway is this more prominent than in the case of auxin, which functions throughout plant development, acting embryonically, post-embryonically, above and below ground. The highly conserved nuclear auxin signal transduction pathway is composed of the TIR1/AFB-Aux/IAA auxin co-receptors, the transcriptional co-repressor TOPLESS (TPL), and the AUXIN RESPONSE FACTORS (ARFs). According to the canonical auxin signaling model, when auxin levels are low Aux/IAAs physically interact with particular ARFs preventing expression of their target genes; in conditions of high auxin however, auxin promotes binding between Aux/IAAs and SCF$^{TIR1/AFB}$ E3 ligases which results in degradation of the transcriptionally repressive Aux/IAAs and allows certain ARFs to activate downstream target genes[1].

Due to their pivotal position in the auxin signaling pathway and expansion in all higher plant species, ARFs are strong candidates for triggering tissue-type and cell type-specific transcriptional changes[2–4]. Phylogenetic analysis has revealed three distinct evolutionarily conserved ARF clades[5]. Based on reporter gene assays, ARFs belonging to clade A are generally considered transcriptional activators, while clade B are repressors, and clade C ARFs show no change in reporter gene expression[6,7]. Although clade A activator ARFs conform to the canonical ARF signaling pathway, the role of the clade B repressor ARFs remains less clear especially given their uncertain interaction with Aux/IAAs[4]. Recent studies suggest that repressor ARFs may compete for binding with activator ARFs to fine tune auxin signaling[3,8].

Despite their proposed functional differences, most ARFs share two conserved structural domains: an N-terminal DNA-binding domain (DBD) and a C-terminal Phox and Bem1p (PB1) domain that mediates homo- and heterodimerization, as well as dimerization with Aux/IAAs which also contain this domain[9–11]. Recent crystallization of the DBD of two Arabidopsis ARFs revealed a dimerization domain within the DBD that allows binding of two adjacent TGTCTC motifs separated by a variable spacer[12]. Spacing between the two repeats was suggested to contribute to ARF binding specificity and has been termed the molecular caliper model. ARFs can also bind as monomers as shown by protein binding microarray (PBM) experiments, but dimerization appears to be relevant for ARF activity and in vivo function[12,13]. Although many gene-scale studies involving the Arabidopsis clade A ARF MONOPTEROS (MP) have identified several direct targets[1,14,15], genome-wide in vivo binding data is available for only a handful of Arabidopsis ARFs[16–18], preventing large-scale comparative analysis of the ARF family.

The maize B73 reference genome contains 33 expressed ARFs[19,20]. No mutant ARF phenotypes have so far been reported in maize, suggesting genetic redundancy and the need for alternative approaches to study ARF function in this important crop where agronomic traits such as plant architecture and drought stress are known to be influenced by auxin[19,21–23]. Here we report genome-wide in vitro DNA binding site maps for fourteen maize ARFs. Overall, our data reveal both specific and redundant aspects of ARF binding that provide a framework for understanding hormone-dependent regulation in species with expanded TF family repertoires. Furthermore, these datasets represent a valuable resource for use in molecular-assisted breeding and genome editing approaches in maize.

## Results

**Genome-wide binding site analysis of maize ARFs.** Using DAP-seq, an in vitro DNA-TF binding assay that captures DNA-binding events in their native sequence context[17], we profiled 26 maize ARFs. Recombinantly expressed ARFs were incubated with maize genomic DNA libraries and ARF-bound DNA fragments were sequenced using next-generation sequencing. Fourteen ARFs produced datasets that passed our stringent quality controls and were further analyzed. These included seven clade A ARFs and seven clade B ARFs, each forming three different sub-clades (Supplementary Fig. 1a; Supplementary Table 1). A genome browser showing binding data is available at https://data.waksman.rutgers.edu/aj2/gallavotti/ZmARFs. ARF datasets were highly reproducible and showed little variation among genomic libraries from different tissues (Supplementary Fig. 1b, c).

In total, 124,530 ARF binding sites were identified (Fig. 1a). Genomic distribution of peaks from all ARF datasets showed enrichment in proximal regulatory regions (Fig. 1b). This enrichment was stronger for clade A ARFs, which showed the majority of their peaks within 10 kb of the TSS (Supplementary Figure 1d, e). Known auxin response genes, such as *Aux/IAAs, GH3s*, and *SAURs* (Supplementary Fig. 2a) showed binding by most ARFs in putative regulatory regions and frequently overlapped with regions of open chromatin identified using an orthogonal ATAC-seq assay (Fig. 1c; see open chromatin profiling section below). We also observed strong peaks in putative regulatory regions of other known targets of Arabidopsis ARFs, including homologs of *TMO6* and *LFY*[14,15], suggesting conserved transcriptional regulation for certain target genes across the >150 million years of evolution that separate Arabidopsis and maize (Supplementary Fig. 2b, 3a–d). Additional genes previously unknown to be directly bound by ARFs included those involved in auxin transport (*BIF2/PINOID/GRMZM2G171822* and *ZmBIG GRAIN/GRMZM2G178852*), auxin degradation (*ZmDAO/GRMZM2G121700*) and signaling (*ZmARF21*) (Supplementary Fig. 4a–c)[23–25]. Strong peaks were also located near hormone pathway genes, such as *ZmDLK2* (strigolactone signaling; *GRMZM2G141999*) and *ZmCKX4* (cytokinin degradation; *GRMZM5G817173*), while others, such as *GRMZM2G121354* (spliceosome-related) and *GRMZM2G109865* (GABA transporter) were uncharacterized (Supplementary Fig. 5a, b)[26,27].

All ARFs preferentially bound to sequences containing the core TGTC motif, however clade A ARFs predominantly showed enrichment for the motif TGTCGG as observed previously for Arabidopsis[12,17], whereas most clade B ARFs showed enrichment for TGTC motifs with a cytosine rich tail (i.e. TGTCCCC, Fig. 1d). Because the latter motif differed from previous reports[12,17,28], we speculated that genomic composition could influence motif enrichment. We therefore performed DAP-seq experiments using maize ARFs incubated with Arabidopsis genomic DNA and found that all were enriched for TGTCGG or TGTC motifs (Supplementary Fig. 5c). In agreement with this finding, removal of reads mapping to maize repetitive regions showed enrichment for the TGTCGG motif (Supplementary Fig. 5d), suggesting that genomic content influences clade B binding.

GO enrichment analysis of high confidence putative target genes showed enrichment for response to various types of hormone stimuli, including auxin and ABA, as well as auxin-mediated signaling components. Additional enrichment was observed for organ development, cell wall related processes, and regulation of gene expression (Fig. 1e). Enrichment terms did not cluster according to clades or sub-clades. Taken together, these results indicate that maize ARF DAP-seq datasets are biologically relevant and constitute an important resource for identifying auxin-regulated genes.

**Binding site specificity among clade A and clade B ARFs.** To assess binding site specificity among the 14 ARFs, we performed comparative analysis and observed that 67–99% of peaks from individual ARF datasets were present in at least one other ARF dataset (Supplementary Table 1). To better understand this relatively low degree of specificity, we ran pairwise Pearson correlation analysis and found that strikingly, all samples strongly clustered according to their clade A or B phylogenetic classification (Fig. 2a). This revealed that these two major ARF clades had distinct binding profiles but that individual members within each major clade shared a considerable number of sites. Indeed, over a quarter of peaks were found in at least half of the clade A and clade B datasets (Supplementary Fig. 6a, b).

We next directly quantified the percentage of overlapping peaks (i.e. peaks with the same genomic coordinates) by calculating a pairwise shared peak matrix using the dataset with the fewest number of peaks as the denominator (Supplementary Fig. 6c). Based on this analysis, ARFs from clade A shared on average ~79% of their peaks while ARFs from clade B shared an average of ~73% of their peaks. This high degree of overlap was particularly obvious in datasets with relatively fewer peaks (i.e ARF18 and ARF13) in which ~90% of peaks were present in another dataset from the same clade (Fig. 2b). ARFs from the same sub-clade tended to share a higher percentage of peaks (average 91% versus an average of 72% for ARFs from different sub-clades) and hierarchical clustering partially recapitulated phylogenetic sub-clades (Supplementary Fig. 6c). For example, ARF4, which forms a sub-clade with ARF29 (96% amino acid identity within the extended DBD; Supplementary Fig. 1a), shared a higher percentage of peaks with ARF29 (83%) relative to ARF16 (68%) with which it shares only 64% amino acid identity (Supplementary Fig. 6d). A similar result was observed for ARF14, which lies in a sub-clade with ARF36 and ARF39, relative to members of the ARF25 sub-clade (Supplementary Fig. 6d).

To further investigate sub-clade-specific binding, we mined our datasets for peaks that were present exclusively in at least one other ARF from the same sub-clade and found that 0.4–29% of peaks within each ARF dataset were likely sub-clade specific (Supplementary Fig. 7a, b; Supplementary Table 1). Motif enrichment analysis of these sub-clade-specific peaks highlighted differential enrichment of nucleotides flanking the core TGTC motif in several cases (Supplementary Fig. 7c). This finding is notable given the crystallographic evidence for direct amino acid: DNA contact at positions outside the core TGTC motif[12,29]. Overall, these results indicate that ARFs from the two major A and B clades bind to a large set of common sites, while also showing some sub-clade specificity.

Our correlation analysis also indicated that while large binding site differences were observed between the two major clades, a subset of sites were shared by both (Fig. 2a). Specifically, using our percent shared peak matrix we found that up to 58% of identified peaks within certain samples were present in both clade A and B datasets (Supplementary Fig. 6c). To determine the overall frequency of shared A and B sites, we created consensus peak sets for all clade A and all clade B ARFs. We examined the overlap between the two consensus datasets and observed 19,603 sites (~16%) that contained at least one clade A and one clade B peak. These peaks, hereafter termed shared peaks (Fig. 1a), were located near diverse genes, including *ZFL1/LFY/ GRMZM2G098813*, *BIF2/PINOID*, and a homolog of the vacuolar auxin transport facilitator *ZmWAT1/GRMZM2G010372* (Supplementary Fig. 2b, 4a and 7d)[30]. We performed motif analysis of clade A-only, clade B-only, and shared peaks and found that while all were enriched for the core TGTC motif, shared peaks resembled clade A-only peaks and lacked the cytosine tail that characterized clade B-only peaks (Fig. 2d). Genomic distribution

analysis revealed that ~50% of clade A-only and shared peaks resided within 10 kb of gene bodies compared to ~30% of clade B-only peaks (Supplementary Fig. 7e).

Overall these results indicate that while ARFs from the two major phylogenetic clades each predominantly bind to a set of clade-specific sites defined by distinctive properties, a portion of binding sites (16%) are bound by both clade A and B ARFs. These shared sites may reflect the high degree of amino acid conservation within and surrounding the B3 DBD. Seven conserved clade-specific amino acid differences are present within this region, including two in the B3 DBD (Supplementary Fig. 8a, b). Further experiments are needed to determine which residues contribute to their unique binding.

**Clade A and clade B ARFs often target the same loci.** We next sought to understand whether the three types of ARF peaks (shared, clade A-only and clade B-only) were associated with unique putative target genes. Cumulatively, we identified 10,322 unique high confidence ARF target genes (peak within 1 kb of TSS), of which 26% contained at least one clade A and one clade B peak (Fig. 2e, Supplementary Fig. 9a, b). Among these, 22% corresponded to shared peaks (Fig. 2e). GO analysis showed that co-occupied genes were enriched for terms such as 'response to auxin', whereas genes exclusively targeted by clade A ARFs were enriched for terms that included 'development' and 'response to stress' (Supplementary Fig. 9c). No enrichment was found for high confidence genes exclusively targeted by clade B ARFs. Because clade B ARF peaks were less frequently observed within 1 kb of the TSS relative to clade A ARFs, but likely still playing a regulatory role, we expanded our locus occupancy analysis to include distal gene targets (defined as the closest gene 10 kb upstream, 3 kb downstream or overlapping the gene body). At this distance, we found that 46% of target genes were co-occupied (Fig. 2e). The additional 20% of co-occupied sites predominantly comprised independent clade A or clade B peaks, not shared peaks. Such situations were exemplified by the *IAA3* locus, which showed a composite arrangement of all three peak types within 6 kb of the gene body (Fig. 2f). These results indicate that clade A and clade B ARFs could frequently co-regulate the same loci either by binding to nearby regions or by directly competing for the same binding site (i.e. shared peaks), either in the same cell or in different tissues.

**ARFs bind adjacent motifs with distinct spacing preferences.** Of the 15 million TGTC sequence matches found in the maize B73v3 genome, less than 0.8% were bound by at least one of the 14 ARFs, indicating that additional features influence binding. ARFs dimerize via two different domains, an N-terminal DBD and a C-terminal PB1 domain[12], both of which have been proposed to stabilize protein and DNA interactions[13,29]. Surface plasma resonance (SPR) experiments have shown that cooperative binding to two adjacent TGTCTC motifs resulted in higher affinity binding relative to monomeric binding[12]. To investigate whether adjacent motifs influenced binding, we tabulated the number of TGTC instances within 100 bp of the peak summit for each ARF dataset. We chose to use the core TGTC motif rather than longer motifs such as TGTCGG identified in this or previous studies[12,17] or the canonical TGTCTC motif[31] due to the low number of peaks that contained two or more of these motifs (i.e. only 1–9% of peaks contained two or more TGTCGGs or TGTCTCs in any orientation; Supplementary Fig. 9d, e). We found that while 13–26% of the peaks for each dataset contained a single TGTC, 55–86% of peaks contained two or more TGTCs (Fig. 3a, Supplementary Fig. 9d, f). In contrast, randomly selected 100 bp genomic regions contained a much higher percentage of

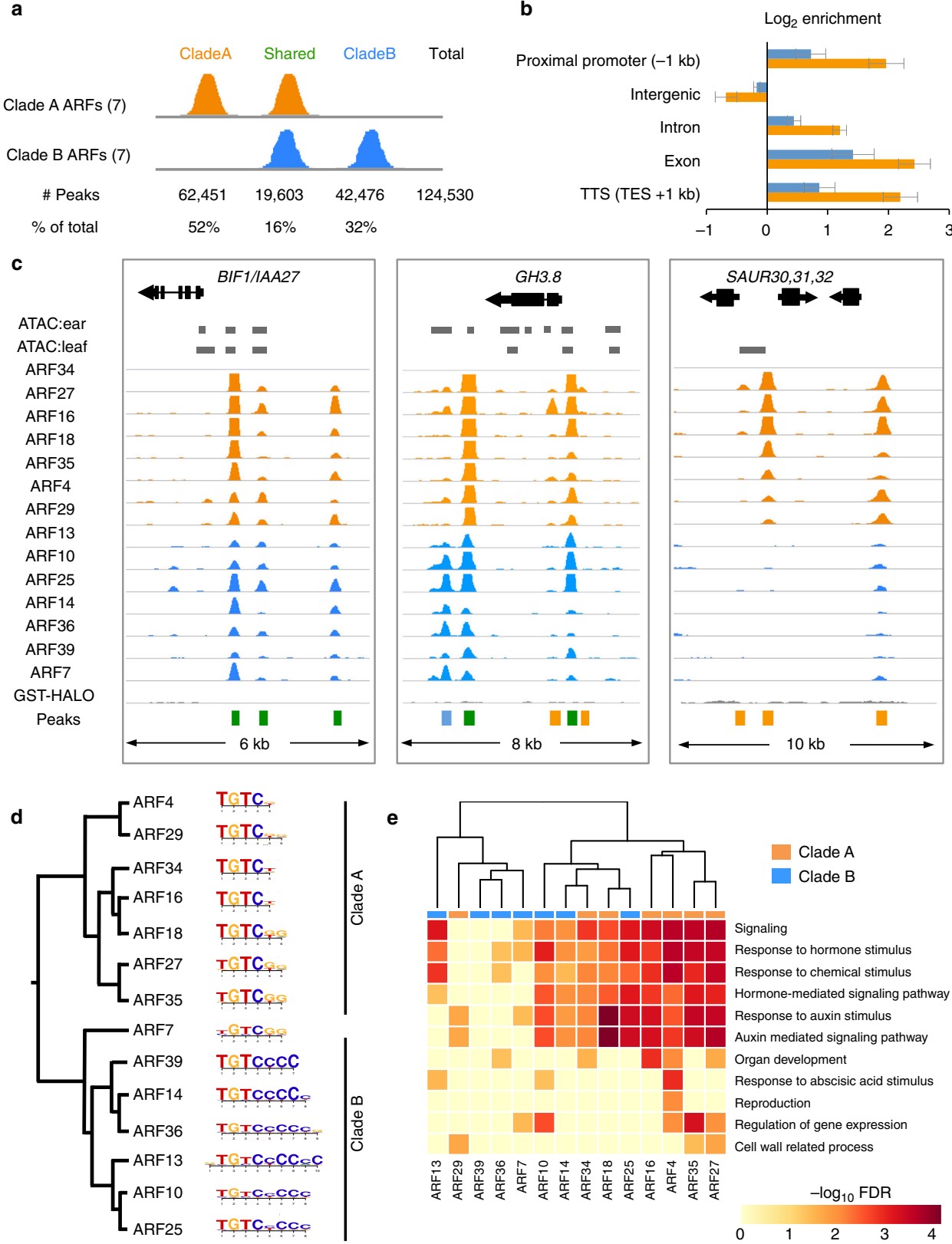

**Fig. 1** ARF binding events are biologically relevant. **a** Total number of clade A (orange) and clade B (blue) ARF peaks identified. **b** Log2 fold enrichment of peaks relative to gene features. Error bars show standard deviation. **c** ARF peaks are located in putative regulatory regions of known auxin responsive genes. GST-HALO track indicates reads from negative control. Regions of accessible chromatin in immature ears and young leaves as determined by ATAC-seq are depicted as gray bars. Colored bars at bottom correspond to called peaks; orange, clade A ARF; blue, clade B ARF; green, peak called in both clade A and B datasets. **d** Top motif identified for each ARF. Dendrogram based on ARF amino acid sequence similarity. **e** Predicted ARF target genes are enriched for functional GO terms related to auxin and other responses

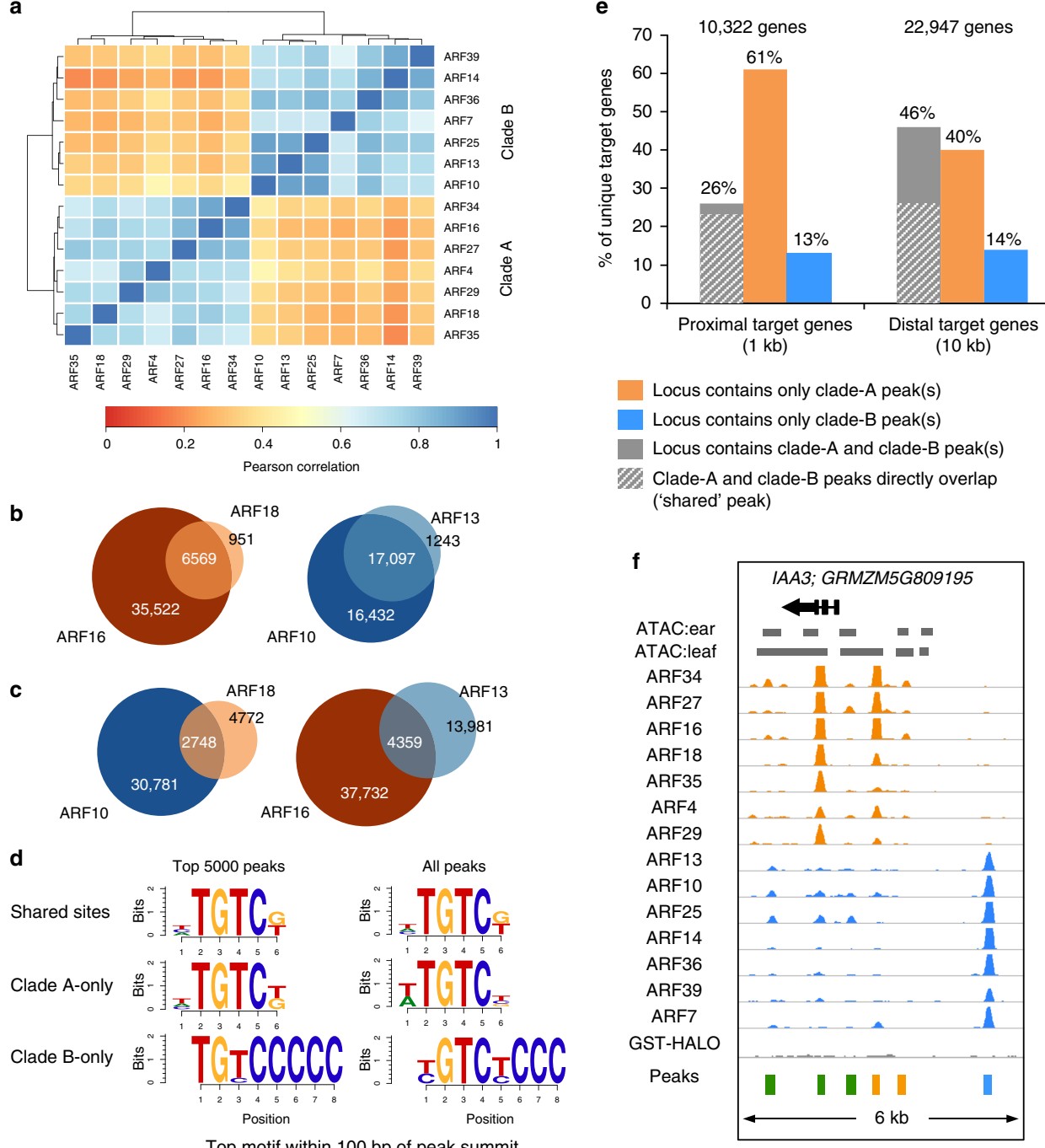

**Fig. 2** Clade A and clade B ARFs bind to unique and shared sites. **a** Heatmap showing Pearson correlation of ARF binding events among each ARF dataset. **b**, **c** Venn diagrams showing peak overlap between representative clade A ARFs (orange) and clade B ARFs (blue). Numeric values show number of peaks. **d** Top motif identified for shared, clade A-only, or clade B-only sites. **e** Percentage of unique proximal (left) and distal (right) putative target genes that contain clade A, clade B, or both peaks. **f** *IAA3* locus targeted by both clade A and B ARFs

instances with zero or only one TGTC (50 and ~34%, respectively), and a much lower percentage of regions with two or more TGTCs (~16%; Fig. 3a and Supplementary Fig. 9f), indicating that ARFs bind more frequently to sites containing multiple TGTCs ($p < 2.2e{-}16$ Fisher's exact test).

To test the hypothesis that high affinity ARF binding is achieved by dimeric binding to adjacent motifs[1,12], we analyzed peak signal intensity relative to TGTC number. Overall, peaks with no TGTCs typically showed the lowest intensities while peaks containing higher numbers of TGTCs showed stronger

intensities, a trend particularly pronounced for clade B ARFs (Fig. 3b, Supplementary Fig. 10a, Methods). These data suggest ARFs preferentially bind to motif clusters, resulting in higher affinity binding sites, or they may reflect the capture of more genomic fragments simply due to the higher density of TGTCs.

We further investigated cooperative ARF binding by examining the orientation and spacing between adjacent TGTCs motifs found on the same peak. ARFs have been shown to bind TGTC-containing repeats in three different orientations with variable numbers of intervening nucleotides (Fig. 3c)[12,17,31]. We therefore

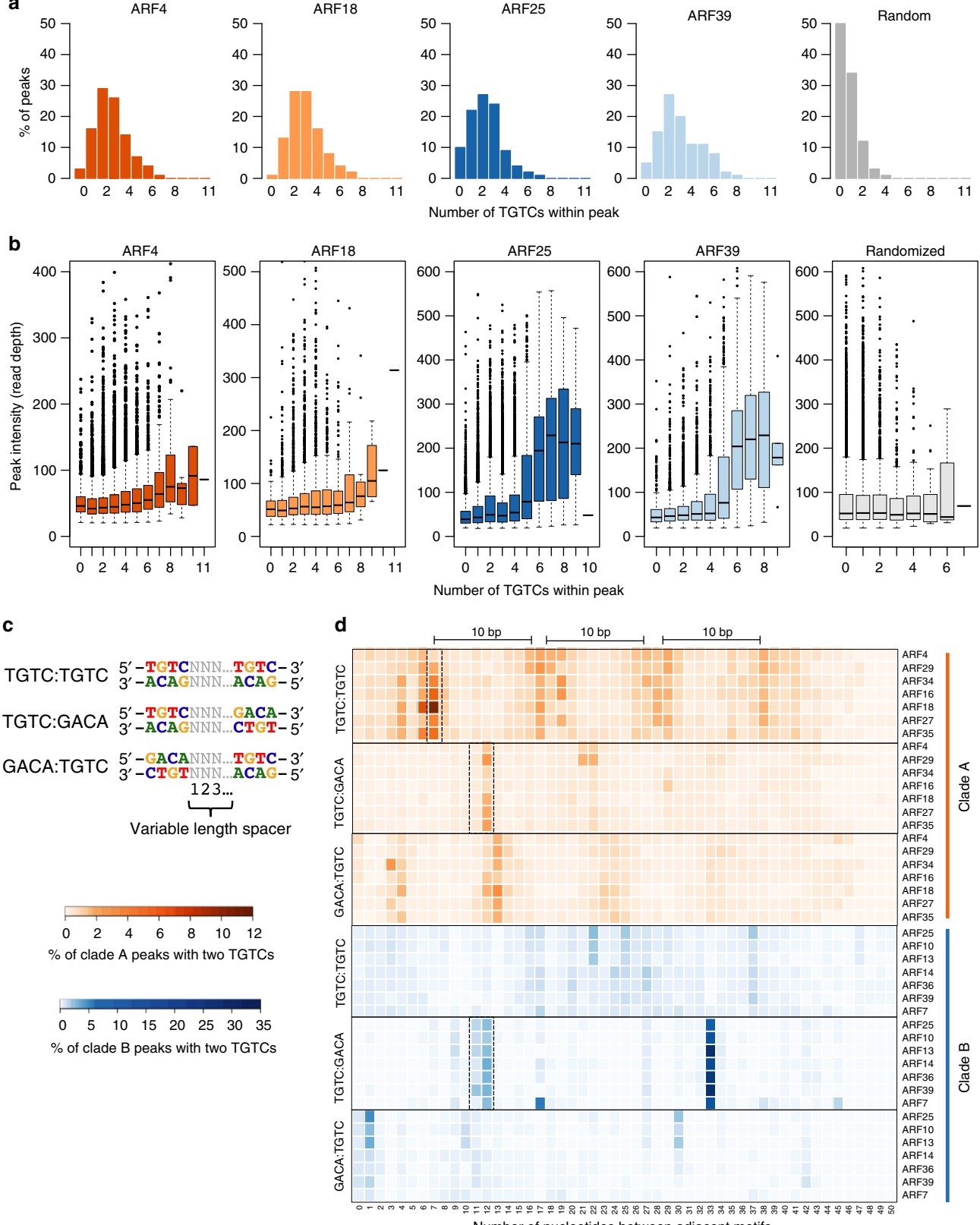

**Fig. 3** ARF peaks frequently contain multiple TGTC repeats. **a** Percentage of total peaks containing different numbers of TGTCs within each peak for four representative ARF datasets and randomly selected regions (gray). **b** Distribution of peak signal intensity (read depth) for peaks containing different numbers of TGTCs. Four representative ARFs and randomly shuffled signal values assigned to random regions (gray) are shown. Central line represents median; upper and lower hinges show first and third quartiles. **c** Schematic showing the three possible TGTC repeat orientations and the numbering system used to describe the number of nucleotides separating the two motifs. **d** Percentage of peaks containing two adjacent TGTCs in the three orientations that contain the indicated number of nucleotides separating adjacent motifs. Black bars highlight 10 bp phasing

extracted all peaks containing two TGTCs and categorized them according to their TGTC:TGTC, TGTC:GACA, or GACA:TGTC orientation. We then tabulated the spacing between the motifs and its frequency. Over 90% of all peaks showed less than 50 intervening nucleotides for all three motif orientations, while 29–46% showed spacing less than 20 nucleotides. While similar clade-specific spacing patterns were observed for most clade A ARFs and most clade B ARFs, substantial differences were observed among the different orientations for each major clade (Fig. 3d; Supplementary Fig. 10b), many of which were independently confirmed by dyad analysis[32] (Supplementary Fig. 10c). These distinct architectural preferences indicate that motif orientation and spacing are both important features that contribute to clade-specific binding site selection. Minor differences were apparent among ARFs of different sub-clades (i.e. ARF4 and 29, TGTC:GACA orientation and ARF7, TGTC:GACA orientation; Fig. 3d).

Notably, our spacing data agreed with structural and SPR data available for the TGTC:GACA orientation (also referred to as everted repeat, ER)[12,31] for AtARF1 (clade B) and AtARF5/MP (clade A) which showed binding to TGTCs separated by 11 and 12 bp (ER7 and ER8)[12], although our clade A ARFs preferred a 12 bp spacer. We also observed preferential binding of clade A ARFs to TGTC:TGTC direct repeat motifs separated by 7 bp, the equivalent of 1xDR5, a commonly used auxin response reporter gene (Fig. 3d, dashed boxes)[31]. Interestingly, preferential binding to TGTC:TGTC orientated motifs by clade A ARFs appeared to show a 10 bp phasing pattern, a distance which corresponds to ~10 bp per helical turn of DNA. Overall, these data indicate that ARFs are able to bind to adjacent motifs separated by multiple, phased helical turns, and that motif architecture is a major binding site determinant for the two major ARF clades, suggesting that sequences outside the highly conserved DBD influence ARF DNA binding. Furthermore, binding to TGTC repeats separated by longer distances suggest that these interactions may occur via the PB1 domain, rather than the DD domain.

**Auxin-induced genes are enriched for proximal ARF binding.** To understand how ARF binding correlated with auxin response, we performed short-term auxin treatment of maize seedlings (100 μM IAA for 30 min) followed by RNA-seq. Differential gene expression analysis identified 339 and 359 genes up- and down-regulated respectively among which 148 were strongly induced (fold change >2; FDR < 0.05; Fig. 4a). GO enrichment analysis of the upregulated genes reported 'auxin-mediated signaling pathway' as the top term and included known early response genes, such as *Aux/IAAs*, *SAURs*, and *GH3s* (Supplementary Fig. 11a, b).

Using the set of auxin upregulated genes, we investigated a number of binding site features that could contribute to auxin inducibility. A small but significant increase in the number of ARF peaks located within 10 kb was observed for auxin-induced genes relative to downregulated or random genes (Fig. 4b; p-value <4e−12, two-sided t-test). This increase was most strongly associated with clade A-only peaks although a significantly higher number of shared peaks were also observed near strongly induced genes relative to randomly selected genes (Supplementary Fig. 11c). We also found that strongly induced genes were over

two times more likely than expected (p-value <2.46e−05 clade A; p-value <0.016 clade B, Fisher's exact test) to have an ARF peak located within 1 kb of the TSS relative to other genes genome-wide (Fig. 4c). This enrichment was seen for both clade A-only and shared peaks (p-value <4.5e−4, Fisher's exact test) but not clade B-only peaks (Fig. 4d). Surprisingly, these data suggest that while robust auxin induction is influenced by the number and location of clade A peaks, many proximal binding sites are also occupied by clade B ARFs (shared peaks).

**Open chromatin profiling reveals tissue-specific ARF binding.** Despite largely overlapping ARF expression patterns (Supplementary Fig. 11d), auxin responses are tissue specific[19,33]. To examine this at the level of DNA binding we sought to identify tissue-specific ARF binding events. Chromatin accessibility is known to impact TF binding[34], and although chromatin status cannot be directly assayed by DAP-seq, integration with open chromatin profiling datasets such as those generated by MNaseHS and ATAC-seq, can provide such information.

We first analyzed open chromatin datasets from four maize tissues: root, seedling/leaf, ear, and tassel determined by either ATAC-seq (this study) or MNaseHS[35]. In agreement with previous DAP-seq findings[17], we observed that in total ~5–25% of ARF peaks overlapped with regions of open chromatin from at least one of these four tissues (Supplementary Fig. 12a, b), with a subset of peaks falling in regions of open chromatin that were unique to each tissue (Fig. 5a). The differences in peak overlap between tissues (i.e. root vs. ear) were proportional to the coverage (total basepairs) of open chromatin regions reported for each tissue dataset (gray shaded areas, Fig. 5a). Overall, clade B ARFs showed a lower percentage of peaks in open chromatin relative to clade A ARFs (Supplementary Fig. 12b, Fig. 5a), and clade B-only peaks were about three times less likely than clade A-only or shared peaks to reside in regions of open chromatin (Fig. 5b). This finding correlated with the overall distal location relative to gene bodies of clade B ARFs (Supplementary Fig. 1e).

We next used the ear-specific peaks to investigate putative *cis*-regulatory regions in a major domestication QTL that regulates the expression of the *TEOSINTE BRANCHED1* (*TB1*) gene located 60 kb upstream of the coding region[36]. Within the proximal component of this QTL controlling ear traits[37], we observed two ARF binding events, one of which corresponded to a shared peak that was located within an ear-specific open chromatin region (Fig. 5c). Given the size of the maize genome and the coverage of ear tissue accessible chromatin regions, this finding is unlikely to occur by chance (p-value <2.2e−16; Fisher's exact test). *TB1* is strongly expressed in ear primordia and differences in ARF binding strength or tissue-specific chromatin dynamics at this region could affect TB1 expression and its influence on modern maize architecture. We analyzed the conservation of peak sequences in maize and teosinte landraces[37] and found broad sequence conservation within and surrounding the TGTC motifs. These findings suggest that ear-specific ARF binding at this region may be an important feature for both maize and teosinte, or alternatively, that despite substantial sequence similarity, differences in DNA methylation and/or chromatin

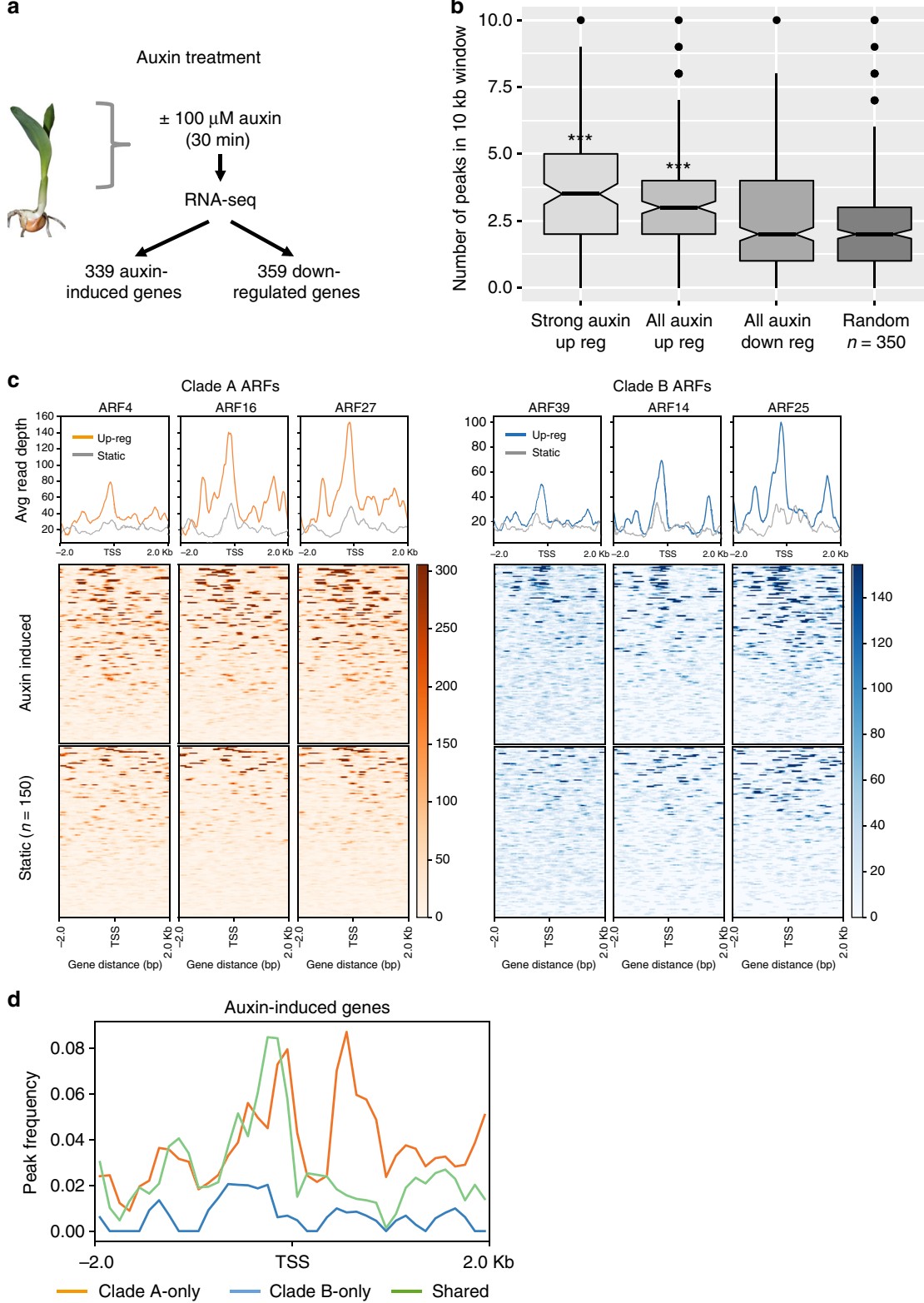

**Fig. 4** Early auxin-induced genes contain ARF peaks proximal to the TSS. **a** Schematic of auxin induction experiment. **b** Auxin-induced genes have a small but significantly greater number of ARF peaks located within 10 kb relative to random genes. *** indicates *p*-value <3.8e−7, pairwise *t*-test; central line shows median, upper, and lower hinges show first and third quartiles, notches show 95% confidence intervals. **c** Representative examples of clade A and clade B ARFs showing increased binding in regions proximal to the TSS for auxin-induced genes relative to a similar number of randomly selected genes. **d** The increase in peak frequency near the TSS of auxin-induced genes can be attributed to sites bound by only clade A ARFs or bound by both clade A and clade B ARFs (shared peaks)

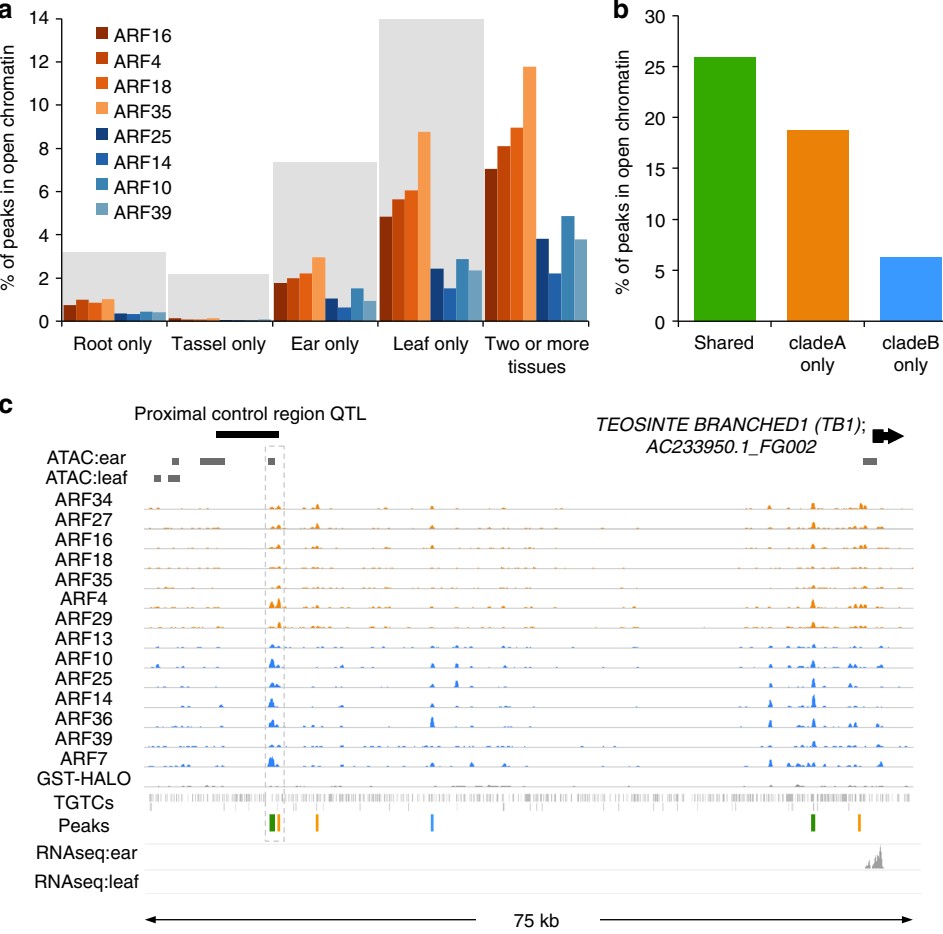

**Fig. 5** ARF peaks overlap with regions of open chromatin. **a** Overlap of DAP-seq peaks with open chromatin regions from different tissues. Gray bars represent coverage of open chromatin datasets relative to leaf dataset (ATAC-seq plus MNase datasets which contained the greatest coverage). Clade A ARFs (orange), clade B ARFs (blue) **b**. Percentage of peaks for the shared, clade A-only, and clade B-only peak types that overlapped with a region of open chromatin found in at least one of four different tissue types. **c** Genome browser screenshot showing overlap of ear-specific region of open chromatin and ARF peaks (gray dashed box) in the proximal control region of the upstream TB1 QTL

accessibility at this region may be present in teosinte and could alter ARF binding.

**Machine learning predicts ARF binding across maize inbreds.** To validate our statistically derived conclusions as well as explore additional non-intuitive sequence features that contribute to ARF binding, we used a supervised machine learning approach to examine TF-DNA binding[38]. Such neural-network-based models have the capacity to capture motif preferences, orientational grammar, and the effects of surrounding sequences on ARF binding[39]. Similar approaches have been widely applied to study regulatory regions of non-coding DNA and have been shown in certain situations to outperform traditional statistical approaches based on explicit starting assumptions[40,41]. We used a subset of each ARF dataset as a training set to develop individual binding models for each ARF. We then asked how good each model was at predicting the binding strength of test sets, which had not been used for training, from every ARF dataset. ARF models were able to clearly distinguish between clade A and B ARFs (Fig. 6a), validating the idea that ARFs within the same clade have more similar binding rules than ARFs from a different clade. For the clade B ARFs, robust models that explained up to 75% of the variation seen in other clade B ARFs were generated and hierarchical clustering based on model predictions recapitulated

phylogenetic groupings (Fig. 6b). Clade A ARF models also predicted binding events among fellow clade A ARFs, but were less predictive for sub-clade specificity (Fig. 6b).

The trained models developed here have the unique power to predict the impact of cis-regulatory variation on ARF binding. We therefore used these models to assess ARF binding potential at a genomic region associated with an herbivore resistance QTL in the maize inbred line Mo17, that in B73 revealed several strong ARF peaks (Fig. 6c)[42]. This region contains a cluster of eight benzoxazinoid (Bx) biosynthesis genes responsible for generating the defense compound DIMBOA. DIMBOA confers resistance to both aphids and European corn borer, two highly destructive insects that affect plant fitness and grain yields[43]. Several of the strongest ARF peaks in this cluster were located downstream of the BX5 gene, in a ~4 kb region called DICE (DIstal Cis-Element) that influences the expression of the BX1 gene, which itself encodes the signature enzyme of the DIMBOA pathway, located about 140 kb downstream (Fig. 6c, Supplementary Fig. 12c)[44].

Mo17 shows strong herbivore resistance and increased BX1 expression relative to B73, phenotypes associated with the partially conserved duplicated DICE element in Mo17 (Fig. 6c, Supplementary Fig. 12d)[42,44]. Application of our machine learning-derived ARF binding models suggested that most

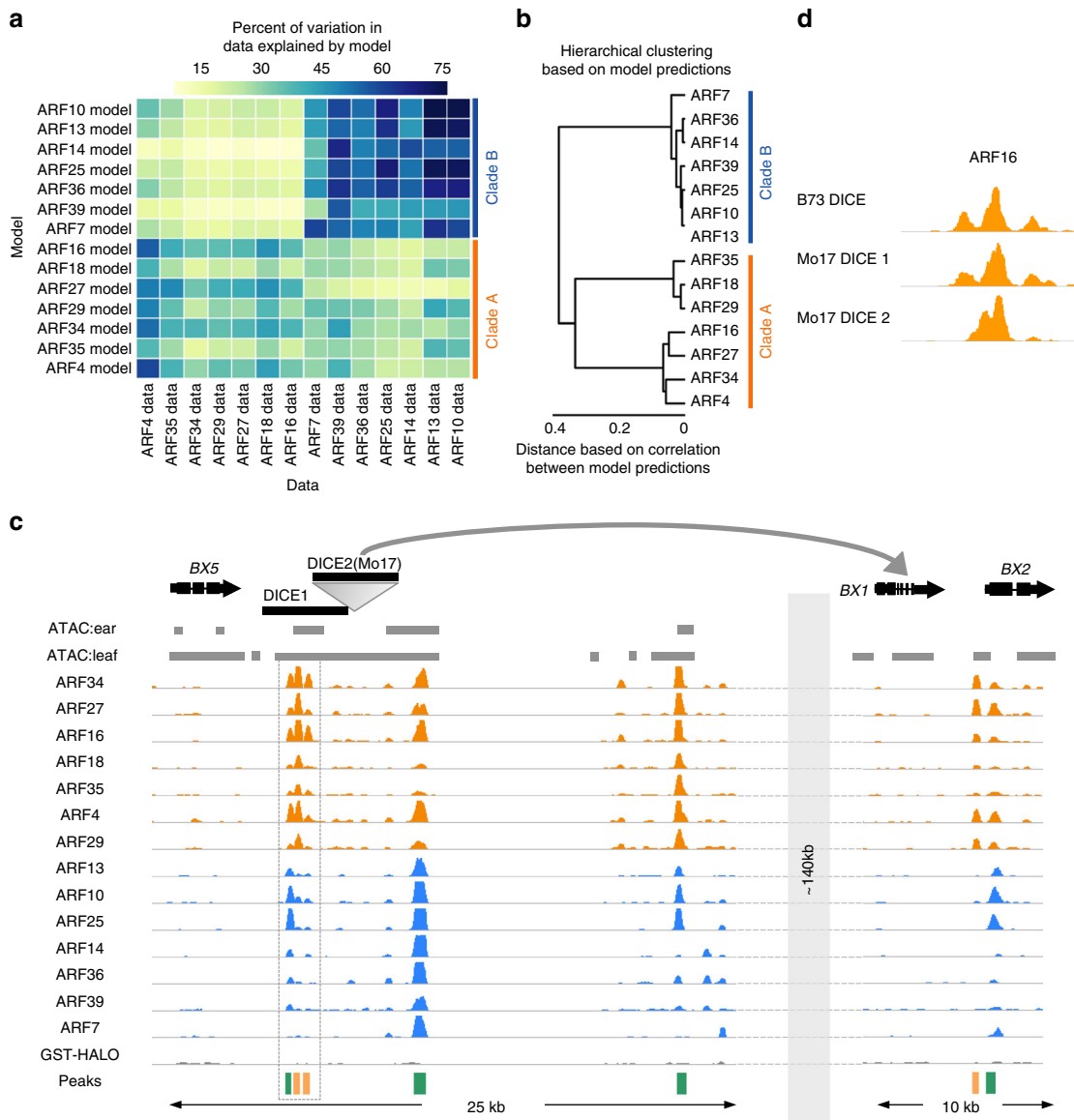

**Fig. 6** ARFs display predictive *cis*-regulatory signatures. **a** Heatmap showing percent of variation in ARF binding data that can be explained by a machine learning model trained on individual ARF datasets. **b** Hierarchical clustering of ARFs based on model predictions. **c** Genome browser view of ARF binding events in the 30 kb region surrounding the DICE element and the *BX1* genic region located 140 kb downstream that is controlled by DICE. **d** Empirically determined ARF16 peaks located in the B73 DICE, Mo17 DICE 1, and Mo17 DICE 2 elements as determined by DAP-seq

clade A ARFs were likely to bind with similar affinity to both the highly conserved first copy of DICE as well as the Mo17-specific second DICE element (Supplementary Fig. 12e, f). We validated the predictive power of these models by performing DAP-seq with Mo17 genomic DNA and ARF16, and observed strong binding in both Mo17 DICE elements (Fig. 6d). Interestingly, ARF16 binding peaks in the second DICE element differed slightly relative to those in the first DICE element, likely due to substantial sequence differences (Fig. 6d, Supplementary Fig. 12d). Overall, both modeling and empirical binding suggest that additional clade A ARF binding sites in Mo17 could be a causative feature that leads to higher *BX1* expression in Mo17. While further experiments are needed to understand the connection between auxin and benzoxazinoid biosynthesis (Zhou et al., 2018), this analysis demonstrates the utility of pairing DAP-seq data with machine learning to explore TF binding across maize inbreds.

## Discussion

TF family expansion is a common feature in many organisms and assessing DNA-binding specificity among different family members will broadly inform our understanding of genetic redundancy and diversification. This study presents large-scale analysis of the ARF family, providing a rich resource of *cis*-regulatory regions controlling many crucial pathways related to the development, domestication, and productivity of maize, an important food source worldwide. Given the high degree of cross-species conservation among ARF family members and target genes, these data also provide a framework to understand numerous aspects of auxin-regulated transcriptional responses.

Surprisingly, we observed relatively few differences in binding site specificity, spacing, and target genes among ARFs from the same clade. This finding suggests that at least for the maize ARFs examined in this study, a high degree of functional redundancy may exist. Such a situation supports the model that the

developmental specificity of auxin-dependent responses may result from ARF binding-induced alterations of chromatin structure that allow different tissue-specific TFs access to nearby cis-elements and thus trigger particular transcriptional programs[45]. This finding does not preclude that despite their relatively infrequent occurrence the several thousand sub-clade-specific sites that were identified (Supplementary Fig. 7a–c) could also be major contributors to certain aspects of auxin response specificity. Tissue-specific ARF expression patterns and genetic analysis in Arabidopsis provide support for both situations[33,46]. Furthermore, ARF heterodimerization and/or interacting partners could result in greater binding site diversity. The combinatorial complexity generated from such interactions could be an additional contributing factor for tissue-specific auxin-directed processes.

As a whole, our genome-wide datasets support a cooperative model of binding by ARF homodimers[12], showing stronger and more frequent binding to sequences containing at least two TGTC motifs. In agreement with the molecular caliper model[12], we observe unique preferential spacing patterns for clade A and B ARFs and show that different preferential spacing occurs for all three possible TGTC repeat orientations at distances spanning up to four helical turns of DNA. We note however that there appear to be other unidentified aspects that contribute to ARF binding since we observed a range of spacing configurations (Fig. 3d). Such factors could include sequences within and surrounding the spacer as demonstrated previously[12,47].

While several clade A Arabidopsis ARFs have been well studied, less is understood about clade B ARFs. Protein interaction data suggest that most clade B ARFs may not interact with Aux/IAAs[8,48] but instead harbor auxin independent functions. A recent study in Physcomitrella, showed that the auxin transcriptional response depends exclusively on Aux/IAAs and proposed that clade B ARFs fine tune auxin signaling by competing for binding with clade A ARF complexes[3]. Our genome-wide data showing that about one third of clade B ARF peaks directly overlap with clade A peaks (Fig. 1a) supports this competitive model. The finding that these shared peaks are frequently located near auxin-induced genes (Fig. 4d), begs the question of how these two clades differ in transcriptional regulatory potential. The presence of the BRD motif on clade B ARFs supports their proposed role as repressors[7,49] and could help to either shut down auxin-induced transcriptional activation by clade A ARFs or prevent spurious activation of sensitive loci in cells where Aux/IAA repressors are not abundant.

Our finding that many clade B-only peaks (~26%; Supplementary Fig. 9b) were located near shared or clade A-only peaks, further supports a model in which clade A and B ARFs cooperate to regulate the same target genes. One caveat to these hypotheses is that because DAP-seq is an in vitro assay, we are unable to assess whether these binding events are co-occurring in the same cell. Given the overlap in many of their expression patterns (Supplementary Fig. 11d), this possibility seems likely. We therefore propose that clade B ARFs are involved in auxin signaling in both a competitive and cooperative fashion with clade A ARFs.

We assessed whether we could derive general rules for ARF transcriptional control[50] and discovered that early auxin responsive genes often contained both clade A and clade B peaks within ~1 kb of the TSS, suggesting the need for tight transcriptional control. We note however that this feature alone was not sufficient to confer auxin inducibility, since we observed many genes with proximal ARF peaks whose expression remained unchanged in our auxin treatment, although their response could simply require different conditions.

We identified >100,000 putative cis-regulatory regions that were directly bound by ARFs and estimated that these binding sites could directly regulate more than a quarter of maize genes. This is comparable to transcriptional analysis of a Physcomitrella aux/iaa triple knockout mutant whereby about one third of genes were mis-regulated[3]. Among putative ARF target loci we identified an herbivore resistance QTL and used neural-network-based models of ARF binding to predict cis-regulatory potential in different maize inbreds. This example illustrates how TF-DNA interaction maps coupled with genetic variation data and genome editing techniques could be used to guide the forward engineering of plant traits to improve crop fitness.

## Methods

**ARF protein expression and purification.** Full length maize pENTR ARFs clones were obtained from Galli et al. 2015 or Grassius[20], except ARF10, ARF12, ARF13 and ARF39 which were cloned from maize inflorescence cDNA into the SfiA and SfiB sites of pENTRsfi using restriction enzyme digestion. Primers used for cloning are shown in Supplementary Table 2. ORFs were transferred into pDEST15 using LR clonaseII to create GST-ARF expression plasmids and transformed into the BL21DE3 codon plus expression strain (Stratagene). Five hundred milliliters cultures of terrific broth with antibiotics was grown to an OD of 0.4–0.8 at 37 °C, induced with 0.4 mM IPTG and grown for 3–4 h at 23 °C. Cell pellets were frozen in liquid nitrogen and stored at −80 °C. Cell pellets were lysed in HBD buffer (25 mM HEPES, 0.7 mM $NA_2HPO_4$, 137 mM NaCL, 5 mM KCl, pH 7.4 with 0.5 mg/ml lysozyme) using sonication. Lysate was cleared by centrifugation for 1 h at 21,000×$g$. Cleared lysate was applied to a GST-gravitrap column (GE Healthcare) washed with HBD buffer, followed by washing with HBDW (20 mM HEPES, 20 mM $Na_2HPO_4$, 137 mM NaCl, 5 mM KCl, 5 mM EDTA, 1% Triton X-100), HBD, and 50 mM HEPES, pH 7.4 + DTT. GST-ARF proteins were eluted from the column using 10 mM reduced glutathione in 50 mM HEPES, pH 8. Proteins were concentrated in Amicon30 Ultracell (MWCO 30) and excess glutathione was removed using two 5 ml buffer exchange steps (50 mM HEPES, pH 7.4). Concentrated proteins were stored in 30% glycerol at −80 °C.

**Illumina library preparation and DAP-seq.** Genomic DNA libraries were prepared according to Bartlett et al. Specifically, gDNA was extracted from maize ear (1–3 cm), tassel (1 cm), and leaf (immature leaves after reproductive transition) using phenol:chloroform:IAA extraction. Five micrograms of genomic DNA was diluted in EB (10 mM Tris-HCl, pH 8.5) and sonicated to 200 bp fragments in a covaris S2 sonicator. DNA was purified using AmpureXP beads at a 2:1 bead:DNA ratio. Samples were then end repaired using the End-It kit (Lucigen) and cleaned using Qiaquick PCR purification (Qiagen) according to the manufacturer's recommendations. Purified samples were A-tailed using Klenow 3–5′exo- for 30 min at RT and then purified using Qiaquick PCR purification as described above. Purified samples were then ligated overnight with a truncated Illumina Y-adapter as described in Bartlett et al. Libraries were purified by bead cleaning using a 1:1 bead:DNA ratio, eluted from the beads in 30 μl of EB, and quantified with the Qubit HS fluorometric assay. A quantity of 20 μl of purified GST-ARF protein (5–20 μg) was diluted in 400 μl of 1X PBS containing 25 μl of washed MagneGST beads (Promega) for 1 h at RT on a rotator to bind the protein to the beads. In addition to the GST-ARF samples, a negative control GST-HALO sample was performed using protein expressed in the TNT wheatgerm expression system (Promega). Beads were washed four times in 1X PBS + NP40 (0.005%) and resuspended in 100 μl of 1X PBS. One microgram of genomic DNA library (from ear tissue except where indicated in Supplementary Fig. 1) was diluted to a final volume of 60 μl in 1X PBS and added to the protein bound beads. Protein bound beads and gDNA were rotated for 1 h at RT. Beads were washed four times in 1X PBS + NP40, followed by two washes with 1X PBS. Beads were transferred to a new tube and DNA was recovered by resuspending in 25 μl EB and boiling for 10 min at 98 °C. Eluted samples were enriched and tagged with dual indexed multiplexing barcodes by performing 20 cycles of PCR in a 50 μl reaction[51]. Samples were pooled and sequenced on a NExtSeq500 with 75 bp single end reads. A total of 10–30 million reads were obtained for each sample.

**Read mapping, filtering, and peak calling.** Fastq files were trimmed using trimmomatic[52] with the following parameters ILLUMINACLIP:TruSeq3-SE:2:30:10 LEADING:3 TRAILING:3 SLIDINGWINDOW:4:15 MINLEN:50. Trimmed reads were mapped to the B73v3 reference genome (nuclear chromosomes only) using bowtie2 v2.2.8[53]. Mapped reads were filtered for reads containing >MAPQ30 using samtools (samtools view −b −q 30) in order to restrict the number of reads mapping to multiple locations in the genome. MAPQ filtered reads were used for all subsequent analysis. Peaks were called using GEM v2.5[54] using the GST-HALO negative control sample for background subtraction and an FDR of 0.00001 (--q 5). Peak calling was performed with the following parameters: --d Read_Distribution_default.txt --k_min 6 --k_max 20 --outNP --sl. ChIPQC v1.8.2 was used to determine FRiP%[55]. Fourteen ARF datasets showed >2% FRiP at the 0.00001 FDR threshold and were used in all subsequent analysis. A blacklist of peak regions appearing in all samples and the HALO-GST negative control was

generated (Supplementary Table 3). Peaks falling in these regions represented less than 0.7% of total peaks and were excluded from all analysis where indicated. For visualization in the Integrative Genome Browser (IGV)[56], bam files were converted to bigwig files using deepTools v2.5.3 bamCoverage with 10 bp bin size and FPKM normalization.

**Technical reproducibility and correlation of ARF datasets.** Scatterplots showing technical reproducibility and tissue variance (Supplementary Fig. 1b, c) were calculated using deepTools multiBamSummary with blacklist subtraction and the following parameters --skipZeros --binSize 200[57]. Pearson correlation (Fig. 2a) was performed on normalized read counts using the multiBigwigSummary and plot-Correlation tools from deepTools. Heatmaps were drawn using the R heatmap functions heatmap.2 and aheatmap.

**Motif enrichment analysis.** The most highly enriched motif for each individual ARF dataset was determined using GEM. All other motifs were determined using meme-chip v4.12.0[58] with the following parameters: -meme-mod anr -meme-minw 4 -meme-maxw 15 -meme-nmotifs 3 -meme-p 8. Fasta sequence files required as input for meme-chip were generated by extracting 50 bp upstream and downstream of the peak summit using the bedtools v2.24.0 slop and getfasta utilities[59]. Motif logos were generated using MotifStack[60].

**Target gene identification and GO analysis.** Gene feature enrichment and peak distances to the nearest gene were assigned using HOMER v4.8.3[61]. Proximal regulatory regions (Fig. 1b) were defined as 1 kb upstream of the transcription start site (TSS) and 1 kb downstream of the transcription termination site (TTS) and gene bodies (exons including 5′- and 3′-UTRs, and introns). Chipseeker[62] was used to determine overlap with gene features. Primary maize B73v3 transcripts were obtained from phytozome (www.phytozome.org) and used for most analysis. For target gene identification used in GO analysis and loci co-occupancy, high confidence target genes were defined as the closest gene containing a peak within 1 kb +100 bp upstream of the TSS and 1 kb −100bp downstream of the TTS (or in the UTR). GO enrichment was performed using AgriGO2.0[63]. The top 15 GO terms with an FDR < 0.05 were selected, and redundant and general terms were collapsed or removed. Meta-gene profiles showing distances relative to the TSS were generated with deepTools using a bin size of 50 bp unless otherwise indicated.

**Identification of open chromatin regions using ATAC-seq.** Ear (1 cm, field grown, frozen tissue), tassel (1 cm, field grown, frozen tissue), and leaf ATAC-seq datasets were generated as described in Lu et al. 2016. Trimmed reads were mapped to the B73v3 reference using bowtie[64] with the following parameters, -X 1000 -v 2 -m 1 --best –strata. PCR duplicates were removed by picard (http://broadinstitute.github.io/picard/) with default parameters. To call high-quality open chromatin regions, (1) pair-end reads were unpaired and changed to single-end reads; (2) primary enriched regions were identified by MACS2[65] with the single-end reads as input; (3) primary enriched regions were split into a series of 50 bp windows with 25 bp overlapping and Tn5 integration events in each window were calculated; (4) windows with Tn5 integration density >=25-fold the average level were picked up and merged together by allowing 150 bp gap to generate the final open chromatin regions.

**Open chromatin and peak overlap analysis.** Root and seedling MNase datasets were obtained from Rodgers-Melnick et al. 2016 and merged with the ear, tassel and leaf ATAC-seq datasets described above using bedops v2.4.26[66] and bedtools v2.24.0[59] for calculations related to the total overlap of ARF peaks with open chromatin. An ARF peak was considered to overlap with an open chromatin region if it overlapped with an ATAC-seq or MNaseHS region by at least one bp. Statistical significance regarding the chance that an ARF peak would be found in the 250 bp ear-specific region of the TB1 proximal QTL (Fig. 5c) was determined by considering the likelihood of overlap between the ear tissue ATAC-seq dataset and each ARF dataset using the bedtools fisher function from bedtools v2.24.0.

Diffbind v2.0.2 occupancy analysis[67] was used to generate the consensus clade A-only, clade B-only and shared peak sets. Bedops and bedtools functions were used to generate the pairwise peak overlap matrix (Supplementary Fig. 6c) and for sub-clade-specific peak analysis (Supplementary Fig. 7c). Peaks were considered overlapping if at least 50% of the peak overlapped in a reciprocal fashion using the –f 0.5 –r options of the bedtools intersect function. The dataset with the fewest number of total peaks for each pairwise comparison was used as the denominator to calculate the percentage of shared peaks. Peaks located in the predetermined blacklist were subtracted for these analyses. To be classified as sub-clade-specific, we required that peaks be found in at least two ARF datasets belonging to the same clade, except in the case of ARF7 which formed a separate clade by itself.

**Cooperative ARF binding and TGTC repeat spacing analysis.** Genome-wide and peak-specific TGTC instances were identified using seqkit[68]. To identify TGTC instances in the ARF peaks, peak sequences were restricted to a 100 bp window surrounding the peak summit. Fasta sequences were extracted from genomic coordinates using bedtools slop and getfasta[59]. Randomized regions were generated

using the bedtools shuffle utility. The spacing between adjacent TGTC motifs shown in Fig. 3d was calculated using peaks containing only two TGTC instances in order to avoid misinterpretation of the motif spacing due to multiple TGTC repeat possibilities. The resulting heatmap scale represents the percentage of the peaks with the indicated spacer length out of the total number of peaks containing two TGTCs. To rule out the possibility that artificial dimerization by the GST-tag could affect ARF DNA binding, we compared data from one clade A and one clade B GST-ARF fusion to in vitro expressed HALOtag-ARF-DBD fusions and found that over 93% of the HALO-tagged ARF peaks were present in the respective GST-tagged ARF datasets. While the HALO-ARF datasets had insufficient %RiP to be considered in our main analysis, we obtained similar results to GST-fusions in terms of motif enrichment, average number of TGTCs within peaks, and spacing patterns.

Peak intensity boxplots and barplots showing read depth in peaks vs. number of TGTCs (Fig. 3b, Supplementary Fig. 10a) were created using the narrowPeak signal value generated by the GEM peak caller. Small but significant increases in clade A peak intensities were observed for peaks containing two TGTCs relative to those with only one TGTC (average 10% increase in signal value among all clade A ARFs, $p \leq 4e-4$ pairwise t-test of least significant ARF dataset). Average signal values increased an additional 15% for peaks containing three or more TGTCs relative to two TGTCs ($p \leq 3e-7$ pairwise t-test of least significant ARF dataset). While a similar trend was observed for the clade B ARFs, most datasets showed a significant increase in peak intensity for peaks containing one TGTC relative to those containing no TGTCs (average 10% increase among all clade B ARFs, $p \leq 0.02$, pairwise t-test of least significant ARF dataset). An additional ~16% average increase in peak intensity was observed for clade B ARF peaks containing two TGTCs relative to one ($p < 0.0186$, pairwise t-test of least significant ARF dataset). Clade B ARF peaks containing three to four TGTCs did not show consistent increases in signal intensity across all datasets, however peaks containing five or more TGTCs displayed substantial increases in signal intensity, with an average increase of 1.5- to 3-fold relative to peaks with only two TGTCs ($p < 2.2e-16$, pairwise t-test of least significant clade B dataset).

**RNA-seq analysis of auxin-induced seedlings.** All aerial portions of 7-day-old maize B73 greenhouse-grown seedlings were collected and incubated in a solution of 100 µM IAA in 1% DMSO or 1% DMSO for 30 min. Samples were snap frozen and total RNA was extracted using RNeasy kit (Qiagen) with DNASe treatment. Three replicates containing three plants each were performed for each treatment. Stranded Illumina RNA sequencing libraries were generated using the TruSeqv3 kit (Illumina) and sequenced on a NextSeq500. Fastq sequences were quality trimmed using trimmomatic as described above and mapped to the B73v3 reference genome using Tophat with default settings[69]. Differential gene expression analysis was performed using cuffdiff[69]. Genes called as differentially expressed with a fold change >2, were considered as strongly induced genes.

Meta-profile plots showing average read coverage in relation to the TSS were generated with deepTools using a bin size of 10 bp. The bedtools window tool was used to calculate the number of ARF peaks present within 10 kb of auxin-induced genes and a similar number of randomly selected genes. GO enrichment of auxin-induced genes was performed using agriGO2.0 and redundant/general terms were collapsed or removed.

**Machine learning analysis.** We utilized a modified version of the DanQ hybrid convolutional-recurrent neural network-based algorithm[38] to perform supervised learning on the ARF DAP-seq data. The training data we used was generated by labeling a sequence 100 basepairs 5′ and 3′ of the called peak location from the B73v3 maize genome with the peak signal strength (read depth). We also selected an equal number of sequences from the genome not in these regions and labeled them with a signal strength of 0 to train the models to recognize what sequences the ARF does not bind to. These 201 bp labeled sequences were then randomly separated into train and test sets for each ARF dataset. All models were trained for 35 cycles and then used for predictions on all the ARF test sets. To calculate the percentage of variability in the data explained by the model we used the pearsonr function from the scipy library[70]. These values were used to generate the heatmap in Fig. 6a using matplotlib and seaborn[71]. The same values were then used to do hierarchical clustering and generate the dendrogram in Fig. 6b.

The trained models were also used to predict relative ARF binding to the Mo17 and B73 DICE elements. We did this by predicting the binding strength of the ARFs to every sequential 201 bp sequence in the DICE element. These data were plotted versus position in Supplementary Fig. 12e using matplotlib and seaborn[71]. The area under each curve in these plots was calculated to give a cumulative ARF binding score to that DICE element and then normalized by the length of the element. This was plotted in Supplementary Fig. 12f using matplotlib and seaborn[71].

## Data availability

Sequence data for experiments performed in this study are available under GEO accession GSE111857. A genome browser displaying mapped reads is available at https://data.waksman.rutgers.edu/aj2/gallavotti/ZmARFs. The python code used to perform

supervised learning on the ARF DAP-seq data can be found at https://github.com/arjunkhakhar/Maize-DapSeq-Machine-Learning.

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

## Acknowledgements

We are grateful to Carol Huang for critical reading of the manuscript and helpful suggestions, Brian Schubert for IT support, and Monika Frey for sharing the Mo17 DICE sequence. M.G. thanks Anna Zedepski, Chris Free, Joe Caracappa, and Yuanchao Zhang for bioinformatics assistance. This research was funded by NSF grant IOS-1546873 to A. G., T.J., and J.N., IOS-1546867 to R.J.S., and a HHMI Faculty Scholar award to J.N.

## Author contributions

M.G. and A.G. conceived the study, analyzed the results, and wrote the manuscript; M.G. performed DAP-seq experiments and bioinformatics analysis; Z.C. performed the auxin induction experiment; A.K. and J.N. wrote and used scripts to analyze the data using machine learning algorithms and contributed to the writing of the manuscript; Z.L and R.J.S. generated the ATAC-seq datasets and contributed to the writing of the manuscript; S.S. and T.J. contributed to bioinformatics analysis.

## Additional information

**Competing interests:** The authors declare no competing interests.

