## [Peer Review File · Nature Communications]

Reviewers' comments:

Reviewer #1 (Remarks to the Author):

This manuscript by Galli and colleagues describes the generation and the analysis of the binding landscape of a large number of Auxin Response Factors (ARFs) in Maize. Auxin is a crucial morphogenetic regulator in plants and ARFs are central to transcriptional regulation in response to auxin. How ARFs contribute to the specificity of auxin responses is an important and open question. The authors use an in vitro high-throughput approach called DAP-Seq to tackle this question and the datasets presented in this manuscript are of broad interest to understand auxin signaling and auxin function during development. The authors also analyzed chromatin opening using ATAC-Seq and analyze the relation between binding and chromatin opening. The analysis presented by the authors is well executed and they consolidate significantly the idea that ARF activators and repressors prefer different motif orientations and spacing (an idea already found in O'Malley et al. 2016). They also show that ARF binding sites are present in important maize QTL. While there are some limited concerns with the analysis of ARF binding (but see below), the significance of binding sites present in regions where QTL are localized (domestication QTL and an herbivore-resistance QTL) is not demonstrated. The paper would benefit from data showing that indeed identifying ARF binding sites in regions where QTL are mapped can help understanding the molecular basis of QTLs.

Major concerns:

1- The authors should make clear from the very start (i.e. abstract, introduction and title of first result section) that the DAP-Seq is an in vitro approach. This is clearly stated quite late in the manuscript, which is quite confusing.

2- P5: it is confusing to introduce already some of the results using ATAC-Seq without any presentation of the approach. The authors should keep that for the dedicated result paragraph and revise/reorganize the figures accordingly.

3- P5: "We also observed strong peaks in putative regulatory regions of other known direct targets of Arabidopsis ARFs including homologs of TMO6 and LFY14,15, suggesting conserved transcriptional regulation across the 150 million years of evolution that separate Arabidopsis and maize (Figure S2B,C)"

It is quite farfetched to use only two ARF targets to support a conclusion on evolutionary conservation. The authors should extend their analysis to other known targets (DRNL, AHP6, etc) and also use the published ARF2/5 DAP-Seq for Arabidopsis (O'Malley et al. 2016) to check whether this is effectively the case. This comparison with the Arabidopsis DAP-Seq datasets will also strengthen the conclusions on the biological relevance of Maize datasets.

4- Beginning of P9: "Only seven conserved class-specific amino acid differences are present within this region and only two are found in the B3 DBD itself (Figure S3J)."

A single amino-acid difference can entirely change the behavior of the protein. It would be interesting to see where these amino-acids are in the structure and whether their localization could suggest a limited impact on the DNA binding. Alternatively the text should be revised.

5- P10 "In contrast, randomly selected 100bp genomic regions contained a much higher percentage of instances with zero or only one TGTC (50% and ~34% respectively), and a much lower percentage of regions with two or more TGTCs (~16%; Figure 3A and Figure S4F), indicating that ARFs bind more frequently to sites containing more than one TGTC motif. "

A statistical analysis would be required to properly support this conclusion.

6- P11 first half of the page: the trend that the authors describe in the text (increase of peak intensity with two TGTC and extra increase with three or more TGTC) is not really seen on Fig 3b where the peak intensity seems to go up more around 7-8 TGTC. Also the authors indicate p-values for significance in the text but nothing is said about the statistical treatment of these data. The authors need to clarify this as it is not clear that their results support their conclusions.

7- P14 on the QTL regulating TB1: it is not clear why the observations reported are significant. First, given the number of binding sites found in the genome, a statistical analysis is needed. Second, domestication has drastically changed plant architecture between teosinte and modern maize and the authors indicate strong conservation of the sites. It is then not clear that such sites would have played a role in domestication and why finding such sites is of interest.

8- From P14 on the use of the machine learning approach: It is not really clear what kind of information the authors really get with this approach notably on the sequences bound by the ARFs. They would need to explain that better so that one can understand why they suddenly use this approach rather than the logo-based analyses they have used before. Also finding ARF binding sites in a herbivore resistance QTL is an interesting correlation but the authors do not provide any evidence that these binding sites are of any functional significance. They authors would need to provide some functional evidence coming for example from existing mutations at these sites or maybe using auxin treatments to show that the sites they find confer different auxin response capacities to the DIMBOA - generating genes in Mo17 and B73.

More minor comments:

9- P3 first paragraph: ARFs are not just activators; "ARFs to activate downstream target genes" needs to be rephrased.

10- P3 "C ARFs show neutral transcriptional activity": I am not sure I know what a neutral transcriptional activity is; this should be rephrased.

11- P14 "we observed that in total ~5-25% of ARF peaks overlapped with regions of open chromatin from at least one of these four tissue types (Figure S1F), with a subset of peaks falling in regions of open chromatin that were unique to each tissue type (Figure 5A, Figure S7A)."

Could the authors describe further whether finding 5-25% of peaks in regions of open chromatin is meaningful? As such these numbers are not very informative.

Reviewer #2 (Remarks to the Author):

The ARF family of transcription factors are central mediators of nearly aspect of plant growth. For decades, members of this family have been characterized as either transcriptional activators or transcriptional repressors, depending on the amino acid composition of the middle region of the protein. Much progress has been made recently on the molecular basis of target gene binding by ARFs, which has led to many new hypotheses on potential mechanisms for ARF target gene specificity. This study fills many of the gaps in our understanding and clarifies what is known about ARF gene targets.

Overall, this manuscript is well-written and clearly laid out and I particularly appreciate the color-coding used throughout the figures. I have no major criticisms of this work, which appears to be well done and thoughtfully analyzed. I have a few minor comments that may improve the manuscript.

Minor comments:

- As an Arabidopsis researcher, I felt that it might be helpful to also have the Arabidopsis ARFs in Figure S1, particularly because roles for these in development have been better characterized than the maize ARF family. Do the authors feel there is a benefit to this, or would this lead to more confusion (i.e., no clear conservation of roles of closely related members)?

- Some of the GEO accession data was missing from the Online Methods section. For example, in the Read Mapping section stating "A blacklist of peak regions appearing in all samples and the HALO -GST negative control was generated (see GEO accession).", no GEO accession number is listed.

- There were a few typos in the Online Methods section, mostly limited to capitalization and space issues. These should be corrected.

Reviewer #3 (Remarks to the Author):

The manuscript by Mary Galli and collaborators reports on the maize DAP-seq exploration of 14 ARF transcription factors (TF), 7 of clade A (activators) and 7 of clade B (repressors). Due to the amount of data and the complexity of the analyses required, this is an important piece of work.

In a nutshell, they obtained their sequence data after cloning and expressing full-length maize ORFs of those TFs fused to Glutathione S-transferase (GST) in order to affinity-capture them bound to genomic DNA fragments. They also tested a negative control protein. As this was done for a collection of cloned TFs one-by-one, it can be assumed that all captured DNA fragments must have been bound by either monomers or homo-oligomers of the same TF.

Regarding the experiment, I have a couple of questions from the point of view of a computational biologist:

- 1) Is there a way to show that the GST fusion is not changing significantly DNA binding, or dimerization, at least for one ARF of each clade? Are there any ChIPseq datasets to compare?
- 2) It would be helpful to see a distribution of length of affinity-captured genomic fragments, so that we know how hard is to discover DNA motifs within them.

Regarding the data analysis, I have more comments/requests:

- 3) On Figure 1 the authors report a large number of DAP-seq peaks (124K). In ChIPseq analysis it is well known that tweaking parameters substantially affects the number of reported peaks, from a few hundreds to hundreds of thousands. Since this number is used all along the paper the authors might want to optimize the related parameters by comparison to known ARF sites, so that precision and recall can be estimated. For instance, they could check whether different parameters change the proportions in barplot Fig1B. Figure 2D suggests this issue is important.
- 4) On page 10 and Figure 3 the authors report on the co-occurrence of TGTC nearby motifs in any orientation. While their analyses make sense, it might be a good idea to confirm those exercises using an algorithm such as dyad-analysis, which is integrated in tool peak-motifs (<https://www.ncbi.nlm.nih.gov/pubmed/18802440> , <https://www.ncbi.nlm.nih.gov/pubmed/27557775>), which rigorously calculates enrichment of this kind of split motifs provided that spacer length is conserved.
- 5) I don't agree that Figure 3B proofs that DNA binding affinity increases with the number of TGTC motifs in a peak. It rather shows that DAP is more efficient (captures more genomic fragments) if several TGTC sites are found.
- 6) The exact phases described in Figure 3D are hard to see. It would be easier if bars of 10bp are added on top of the heatmap, at least for clade A. For both clades it would be helpful to have conceptual models/plots of how dimers bind to those sites, if the A. thaliana protein structures allow it.
- 7) Regarding Figure 4B, isn't 10 kb too wide a distance interval, particularly since gene bodies and

proximal regions are enriched? What happens if 1kb is used instead? This is related to Fig4C. In either case, the barplot would improve if notches were added to define, for instance, 95% confidence intervals around the medians.

8) I have several comments about the machine learning section:

8.1) It's not clear the size of the training and validation sequences sets, nor whether training and validation sets were made by separate. In addition, the code should be published in a source code repo for the sake of transparency and reproducibility.

"The python code to implement this is available upon request" is not good enough because the code could change in the future with no logged changes.

8.2) The authors say that "ARF16 binding peaks in the second DICE element differed slightly relative to those in the first DICE element, likely due to substantial sequence differences (Figure 6D)." Instead of this the authors should report objectively on the exact sequence differences at that locus.

9) On page 4 the authors say that "Maize contains 32 expressed ARFs". In the current context of pangenomes and pantranscriptomes

(<http://www.plantcell.org/content/early/2014/0131/tpc.113.119982>)

the authors should say to which cultivar(s) this figure applies.

is it B73?

We would like to thank the reviewers for their comments and suggestions that we feel overall significantly improve and strengthen our manuscript. We addressed all of the concerns raised and adjusted the manuscript text and Figures accordingly. All changes have been highlighted in the revised text.

Reviewers' comments:

Reviewer #1 (Remarks to the Author):

This manuscript by Galli and colleagues describes the generation and the analysis of the binding landscape of a large number of Auxin Response Factors (ARFs) in Maize. Auxin is a crucial morphogenetic regulator in plants and ARFs are central to transcriptional regulation in response to auxin. How ARFs contribute to the specificity of auxin responses is an important and open question. The authors use an *in vitro* high-throughput approach called DAP-Seq to tackle this question and the datasets presented in this manuscript are of broad interest to understand auxin signaling and auxin function during development. The authors also analyzed chromatin opening using ATAC-Seq and analyze the relation between binding and chromatin opening. The analysis presented by the authors is well executed and they consolidate significantly the idea that ARF activators and repressors prefer different motif orientations and spacing (an idea already found in O'Malley et al. 2016). They also show that ARF binding sites are present in important maize QTL. While there are some limited concerns with the analysis of ARF binding (but see below), the significance of binding sites present in regions where QTL are localized (domestication QTL and an herbivore-resistance QTL) is not demonstrated. The paper would benefit from data showing that indeed identifying ARF binding sites in regions where QTL are mapped can help understanding the molecular basis of QTLs.

Major concerns:

1- The authors should make clear from the very start (i.e. abstract, introduction and title of first result section) that the DAP-Seq is an *in vitro* approach. This is clearly stated quite late in the manuscript, which is quite confusing.

This was an unintentional oversight. We have modified the text of the abstract, intro and first results as follows:

ABSTRACT

“We used DAP-seq to generate genome-wide *in vitro* TF:DNA interaction maps for fourteen maize ARFs

INTRODUCTION

“Here we report genome-wide *in vitro* DNA binding site maps for fourteen maize ARFs.”

RESULTS

“Using DAP-seq, an *in vitro* DNA-TF binding assay...”

2- P5: it is confusing to introduce already some of the results using ATAC-Seq without any presentation of the approach. The authors should keep that for the dedicated result paragraph and revise/reorganize the figures accordingly.

We have moved Fig. S1f to Fig. S7b which describes other aspects of the open chromatin overlap. However, we kept the reference to the ATAC-seq assay on page 5 because in all the figures (see Fig 1c, 2f, Figure S2b,c,d etc.) we present before the open chromatin results section, ATAC-seq regions are shown. These regions strengthen the validity of the binding sites in all these figures. To address potential confusion, we have modified the text as follows:

“...and frequently overlap with regions of open chromatin identified using an orthogonal ATAC-seq assay (Figure 1c, S2a; see open chromatin profiling section below).”

3- P5: "We also observed strong peaks in putative regulatory regions of other known direct targets of Arabidopsis ARFs including homologs of TMO6 and LFY14,15, suggesting conserved transcriptional regulation across the 150 million years of evolution that separate Arabidopsis and maize (Figure S2B,C)"

It is quite farfetched to use only two ARF targets to support a conclusion on evolutionary conservation. The authors should extend their analysis to other known targets (DRNL, AHP6, etc) and also use the published ARF2/5 DAP-Seq for Arabidopsis (O'Malley et al. 2016) to check whether this is effectively the case. This comparison with the Arabidopsis DAP-Seq datasets will also strengthen the conclusions on the biological relevance of Maize datasets.

We appreciate the reviewer pointing out this potentially misleading statement. Our observation about the conservation of two conserved target genes was not intended to imply that there was a high degree of conservation among all Arabidopsis and maize target genes. We have modified the text to clarify our statement that *some* ARF target genes are likely conserved across evolution. We have modified the text as follows:

“suggesting conserved transcriptional regulation for certain homologous target genes across the 150 million years of evolution that separate Arabidopsis and maize (Figure S2B-F)”. To strengthen this statement we have examined ARF binding in several more maize homologs of known Arabidopsis targets (DRNL, AHP6, and TMO7) as suggested by the reviewer. These have been added as additional panels to the supplemental figures (Fig. S2D-F). We do not include an analysis of Arabidopsis ARF binding events because we feel that this distracts from the main focus of our paper, which is to compare different *maize* ARF binding events. We also wish to note that this type of comparative analysis is complex given the duplicated nature of the maize genome. Even co-orthologous genes in maize contain different regulatory regions. For example, we see ARF peaks in the ZFL1 but not ZFL2 gene despite both being highly orthologous to AtLFY.

4- Beginning of P9: "Only seven conserved class-specific amino acid differences are present within this region and only two are found in the B3 DBD itself (Figure S3J)."

A single amino-acid difference can entirely change the behavior of the protein. It would be interesting to see where these amino-acids are in the structure and whether their localization could suggest a limited impact on the DNA binding. Alternatively the text should be revised.

We have modified the text to account for the reviewer's comment. We chose to modify the text instead of including an analysis of where these residues are on the protein structure, because we feel that such analysis can be more thoroughly addressed in future studies with experimental validation. The new text reads: "Seven conserved class-specific amino acid differences are present within this region, including two that are found in the B3 DBD itself"

5- P10 "In contrast, randomly selected 100bp genomic regions contained a much higher percentage of instances with zero or only one TGTC (50% and ~34% respectively), and a much lower percentage of regions with two or more TGTCs (~16%; Figure 3A and Figure S4F), indicating that ARFs bind more frequently to sites containing more than one TGTC motif. "

A statistical analysis would be required to properly support this conclusion.

We have corrected this unintentional omission. The text now reads:

"In contrast, randomly selected 100bp genomic regions contained a much higher percentage of instances with zero or only one TGTC (50% and ~34% respectively), and a much lower percentage of regions with two or more TGTCs (~16%; Figure 3A and Figure S4F), indicating that ARFs bind more frequently to sites containing more than one TGTC motif ($p < 2.2e-16$ Fisher's exact test)."

6- P11 first half of the page: the trend that the authors describe in the text (increase of peak intensity with two TGTC and extra increase with three or more TGTC) is not really seen on Fig 3b where the peak intensity seems to go up more around 7-8 TGTC. Also the authors indicate p-values for significance in the text but nothing is said about the statistical treatment of these data. The authors need to clarify this as it is not clear that their results support their conclusions.

We have clarified the text to address the reviewer's concern. The text now reads:

"Small but significant increases in clade A peak intensities were observed for peaks containing two TGTCs relative to those with only one TGTC (average 10% increase in signal value among all clade A ARFs, $p \leq 4e-4$ pairwise t-test of least significant ARF dataset). Average signal values increased an additional 15% for peaks containing three or more TGTCs relative to two TGTCs ($p \leq 3e-7$ pairwise t-test of least significant ARF dataset). While a similar trend was observed for the clade B ARFs, most datasets showed a significant increase in peak intensity for

peaks containing one TGTC relative to those containing no TGTCs (average 10% increase among all clade B ARFs, $p \leq 0.02$, pairwise t-test of least significant ARF dataset). An additional ~16% average increase in peak intensity was observed for clade B ARF peaks containing two TGTCs relative to one ($p < 0.186$, pairwise t-test of least significant ARF dataset). Clade B ARF peaks containing three to four TGTCs did not show consistent increases in signal intensity across all datasets, however peaks containing five or more TGTCs displayed substantial increases in signal intensity, with an average increase of 1.5- to 3-fold relative to peaks with only two TGTCs ($p < 2.2e-16$, pairwise t-test of least significant clade B dataset). Taken together these data support a model in which ARFs preferentially bind to motif clusters, resulting in higher affinity binding sites.”

7- P14 on the QTL regulating TB1: it is not clear why the observations reported are significant. First, given the number of binding sites found in the genome, a statistical analysis is needed. Second, domestication has drastically changed plant architecture between teosinte and modern maize and the authors indicate strong conservation of the sites. It is then not clear that such sites would have played a role in domestication and why finding such sites is of interest.

To address the first concern, we now provide statistical support showing that given an effective genome size of ~2Gb (nuclear chromosomes), the chances that one of our ARF peaks lies in a region of ear open chromatin is unlikely to occur by random chance. We have modified the main text and methods as follows:

Main text:

“Within the proximal component of this QTL controlling ear traits ⁴¹, we observed two ARF binding events, one of which corresponded to a shared peak occupied by both clade A and B ARFs located within an ear-specific open chromatin region (Figure 5C). Given the size of the maize genome and the coverage of ear-tissue accessible chromatin regions, this finding is unlikely to occur by chance (p -value $< 2.2e-16$; Fisher’s exact test).”

METHODS

“Statistical significance regarding the chance that an ARF peak would be found in the 250bp ear-specific region of the TB1 proximal QTL (Figure 5C) was determined by considering the likelihood of overlap between the ear tissue ATAC-seq dataset and each ARF dataset using the bedtools fisher function from bedtools v2.24.0.”

Regarding the reviewer’s second comment that we observe no difference at the DNA sequence level in the ARF binding site between maize and teosinte: Despite this high degree of sequence homology, we feel that ARF binding events located in this region are of interest for at least two reasons. First, regardless of whether these binding events occur in teosinte, our finding that ARF peaks are present in an ear-specific region of open chromatin in modern maize is interesting in and of itself for those studying tissue-specific TF binding dynamics

and maize inflorescence architecture. Second, a high degree of DNA sequence conservation among maize and teosinte does not eliminate the possibility that there could be DNA methylation differences and/or differences in tissue-specific open chromatin within the teosinte sequence that could affect ARF binding. Because such factors can influence TF binding, we feel that the presence of the ARF binding site will be of interest to people investigating this particular QTL.

We have modified the text to hopefully better clarify these possibilities:

“We analyzed the conservation of peak sequences in different maize and teosinte landraces ⁴¹ and found a high degree of sequence conservation within and surrounding the TGTC motifs. These findings suggest that ear-specific ARF binding at this region may be an important feature for both sub-species, or alternatively, that despite the high degree of sequence similarity, differences in DNA methylation and/or chromatin accessibility at this region may be present in teosinte and could alter ARF binding.”

8- From P14 on the use of the machine learning approach: It is not really clear what kind of information the authors really get with this approach notably on the sequences bound by the ARFs. They would need to explain that better so that one can understand why they suddenly use this approach rather than the logo-based analyses they have used before. Also finding ARF binding sites in a herbivore resistance QTL is an interesting correlation but the authors do not provide any evidence that these binding sites are of any functional significance. They authors would need to provide some functional evidence coming for example from existing mutations at these sites or maybe using auxin treatments to show that the sites they find confer different auxin response capacities to the DIMBOA -generating genes in Mo17 and B73.

We would like to thank the reviewer for pointing out the lack of clarity in our explanation of the utility of the machine learning approach and the models generated therein. Our machine learning approach was first intended to serve as an unbiased method to assess ARF binding trends. Indeed the independent models generated by this method supported our findings regarding the differences and similarities among clade A and clade B ARFs. We then chose to demonstrate how this machine learning approach could be used predict TF binding across maize inbred lines. Certainly we could have run 14 DAP-seq assays with each of our ARFs using Mo17 DNA, however we intended to show that a machine learning approach could be used as a scalable, cost-saving measure to instead predict how binding would be affected in these lines. The empirical binding to the Mo17 DICE element that we observed with ARF16 supported the machine learning findings and demonstrated that our predictions for the other thirteen ARF datasets were reliable. We envision that such an approach could ultimately be used to predict large scale TF binding events across many maize inbred lines and serve as a basis to investigate cis-regulatory differences. We have attempted to more explicitly state these points by editing

the text of both the results and discussion section.

RESULTS

“To validate the statistically-derived conclusions regarding sequence features that contribute to ARF binding and to explore the possible existence of additional non-intuitive features that govern ARF binding, we used a supervised machine learning approach previously developed for examining TF DNA binding⁴⁵. These neural-network based models have the capacity to capture binding motif preferences, orientational grammar, and the effects of surrounding sequences on ARF binding⁴⁶. Similar machine-learning based approaches have been widely applied to study the regulatory regions of non-coding DNA and have been shown in certain situations to outperform traditional statistical approaches that set out to test explicit starting assumptions^{47,48}. We used a subset of each ARF dataset as a training set to develop individual binding models for each ARF. We then asked how good each model was at predicting binding strength of test sets, which had not been used for training, from every ARF dataset. ARF models were able to clearly distinguish between clade A and clade B ARFs (Figure 6A), validating the idea that ARFs within a clade have binding rules more similar to each other than to ARFs from a different clade.”

....

“In addition to adding credence to the conclusions arrived at through other means of analysis, the trained models developed here have a unique power: to predict the impact of cis-regulatory variation on ARF binding. We therefore used these models to assess ARF binding potential at a genomic region associated with an herbivore resistance QTL in the maize inbred line Mo17, that in B73 revealed several strong ARF peaks (Figure 6C).”

DISCUSSION

“With the emergence of genome editing techniques, tools such as the predictive models developed here could be used for forward engineering of plant traits to revolutionize plant breeding approaches.”

The second concern raised by the reviewer regards providing evidence for the function of the additional ARF binding sites in Mo17. Unfortunately, we are not aware of any existing material containing mutations at these sites that could be used to support their functionality. Targeted mutagenesis of these binding sites in Mo17 would be an excellent follow up experiment and we are certainly interested in testing this, although it is a long-term goal given the time needed for maize transformation. We also like the proposed idea to carry out auxin treatments on B73 and Mo17, however we feel that thoroughly substantiating a role for auxin in the BX biosynthesis pathway would require extensive experiments and functional data that would distract from the central focus of this manuscript. We do note however that while no direct connection has yet been made between BX biosynthesis and auxin signaling, there is some evidence suggesting that benzoxazinoids could have a role in auxin signaling. We have included a reference to the paper where this is discussed and have added further language

to the conclusion of that section to make it clear we are not claiming a causal link. The purpose of that experiment was to demonstrate the predictive power of the models that we had developed and show how they could be used to guide functional studies, which we feel we have done adequately.

We have modified the text as follows:

“Overall, both modeling and empirical binding results suggest that additional clade A ARF binding sites in Mo17 could be a causative feature that leads to higher *BX1* expression in Mo17. While further experiments are needed to better understand the incipient connection between auxin and benzoxazinoid biosynthesis (Zhou et al., 2018), this analysis demonstrates the utility of pairing DAP-seq data with machine learning to explore TF binding across maize inbred lines.”

More minor comments:

9- P3 first paragraph: ARFs are not just activators; "ARFs to activate downstream target genes" needs to be rephrased.

We intended here to highlight how the canonical auxin signaling pathway is believed to operate. We have modified the text to account for the fact that not all ARFs fit into this pathway.

“According to the canonical auxin signaling model, when auxin levels are low Aux/IAAs physically interact with particular ARFs preventing expression of their target genes; in conditions of high auxin however, auxin promotes binding between Aux/IAAs and SCF^{TIR1/AFB} E3 ligases which results in degradation of the transcriptionally repressive Aux/IAAs and allows certain ARFs to activate downstream target genes¹.”

10- P3 "C ARFs show neutral transcriptional activity": I am not sure I know what a neutral transcriptional activity is; this should be rephrased.

The text has been changed to:

“Based on reporter gene assays, ARFs belonging to clade A are generally considered transcriptional activators, while clade B are repressors, and clade C ARFs show no change in reporter gene expression⁵⁻⁷.”

11- P14 "we observed that in total ~5-25% of ARF peaks overlapped with regions of open chromatin from at least one of these four tissue types (Figure S1F), with a subset of peaks falling in regions of open chromatin that were unique to each tissue type (Figure 5A, Figure S7A)."

Could the authors describe further whether finding 5-25% of peaks in regions of open chromatin is meaningful? As such these numbers are not very informative. As stated in the text (“In agreement with previous DAP-seq finding, we observed that in total ~5-25% of ARF peaks overlapped with regions of open chromatin...”), we have compared our ARF peak and open chromatin overlap

data with that reported previously for three different Arabidopsis TFs. O'Malley et al., 2015 found that ~30-45% of DAP-seq peaks overlapped with DHS sites from different tissues. Our results showing slightly lower percentages of overlap relative to those reported in O'Malley et al, likely reflect differences in open chromatin profiling dataset depth or stringency and/or differences in TF behavior. The later is likely the case for the clade B ARFs, which as noted in the text showed an overall lower overlap with regions of open chromatin relative to clade A ARFs.

Reviewer #2 (Remarks to the Author):

The ARF family of transcription factors are central mediators of nearly aspect of plant growth. For decades, members of this family have been characterized as either transcriptional activators or transcriptional repressors, depending on the amino acid composition of the middle region of the protein. Much progress has been made recently on the molecular basis of target gene binding by ARFs, which has led to many new hypotheses on potential mechanisms for ARF target gene specificity. This study fills many of the gaps in our understanding and clarifies what is known about ARF gene targets.

Overall, this manuscript is well-written and clearly laid out and I particularly appreciate the color-coding used throughout the figures. I have no major criticisms of this work, which appears to be well done and thoughtfully analyzed. I have a few minor comments that may improve the manuscript.

Minor comments:

- As an Arabidopsis researcher, I felt that it might be helpful to also have the Arabidopsis ARFs in Figure S1, particularly because roles for these in development have been better characterized than the maize ARF family. Do the authors feel there is a benefit to this, or would this lead to more confusion (i.e., no clear conservation of roles of closely related members)?

We feel this is an excellent addition and have modified the phylogeny in Figure S1.

- Some of the GEO accession data was missing from the Online Methods section. For example, in the Read Mapping section stating "A blacklist of peak regions appearing in all samples and the HALO-GST negative control was generated (see GEO accession).", no GEO accession number is listed. We appreciate the reviewer catching this error. We were unable to submit this data to GEO due to their submission criteria. We instead had attached this as a supplemental table, but forgot to update the Online methods section. We have now modified the methods to read:

"A blacklist of peak regions appearing in all samples and the HALO-GST

negative control was generated (Supplemental table S2).”

- There were a few typos in the Online Methods section, mostly limited to capitalization and space issues. These should be corrected.

We proofread the methods section and corrected many of errors. Please see track changes in text document.

Reviewer #3 (Remarks to the Author):

The manuscript by Mary Galli and collaborators reports on the maize DAP-seq exploration of 14 ARF transcription factors (TF), 7 of clade A (activators) and 7 of clade B (repressors). Due to the amount of data and the complexity of the analyses required, this is an important piece of work.

In a nutshell, they obtained their sequence data after cloning and expressing full-length maize ORFs of those TFs fused to Glutathione S-transferase (GST) in order to affinity-capture them bound to genomic DNA fragments. They also tested a negative control protein. As this was done for a collection of cloned TFs one-by-one, it can be assumed that all captured DNA fragments must have been bound by either monomers or homo-oligomers of the same TF.

Regarding the experiment, I have a couple of questions from the point of view of a computational biologist:

1) Is there a way to show that the GST fusion is not changing significantly DNA binding, or dimerization, at least for one ARF of each clade? Are there any ChIPseq datasets to compare?

Unfortunately, there are no maize ChIPseq datasets that we can compare to. We have however performed a DAP-seq experiment using the DNA-binding domain (DBD) of ARF4 (cladeA) and ARF39 (cladeB) fused to a HALO-tag and expressed in an in vitro rabbit reticulocyte-based transcription/translation system. (Full length HALO-ARF constructs were also tested but failed to produce any peaks). While these HALO-ARF-DBD experiments returned fewer peaks (below our cutoff for success datasets) than our E.coli purified GST-ARF DAP-seq experiments (possibly due to a lower amount of protein), over 93% of the HALO-tagged ARF peaks were present in the GST-tagged ARF datasets. Motif enrichment produced identical top motifs for HALO-DBD fusions, no statistical difference was seen between the average number of TGTCs within peaks (Fisher's exact test), and spacing patterns were similar. We therefore believe that the GST tag likely does not influence ARF binding activity.

2) It would be helpful to see a distribution of length of affinity-captured genomic fragments, so that we know how hard is to discover DNA motifs within them.

As the first step in our library construction we have sheared our genomic DNA to 200bp fragments using a covaris S2 followed by an Ampure bead size selection step. This procedure produces very well defined genomic fragments of the expected size. When viewed by agarose gel separation, a typical final library which includes illumina sequencing adapters produces fragments of 350-400bp. Given that the illumina sequencing adapter adds ~170 bp total, we note that most input genomic fragments are ~200bp in length.

Regarding the data analysis, I have more comments/requests:

3) On Figure 1 the authors report a large number of DAP-seq peaks (124K). In ChIPseq analysis it is well known that tweaking parameters substantially affects the number of reported peaks, from a few hundreds to hundreds of thousands. Since this number is used all along the paper the authors might want to optimize the related parameters by comparison to known ARF sites, so that precision and recall can be estimated. For instance, they could check whether different parameters change the proportions in barplot Fig1B. Figure 2D suggests this issue is important.

Unfortunately, there are no known maize ARF sites with which to calibrate our cutoffs. We initially tested our data however with three different FDR thresholds (FDR<0.01, 1e-5, and 1e-7) and chose a universal cutoff of FDR< 1e-5 which maximized the number of datasets giving greater than 2% reads in peaks (RiP) while restricting peak numbers such that comparative analysis between ARF datasets was reasonable. At each cutoff level, similar numbers of gene feature enrichments (Fig.1B) were observed, although clade A ARFs run with default parameters produced slightly decreased enrichment in promoter, TTS and exonic peaks.

4) On page 10 and Figure 3 the authors report on the co-occurrence of TGTC nearby motifs in any orientation. While their analyses make sense, it might be a good idea to confirm those exercises using an algorithm such as dyad-analysis, which is integrated in tool peak-motifs

(<https://www.ncbi.nlm.nih.gov/pubmed/18802440> ,

<https://www.ncbi.nlm.nih.gov/pubmed/27557775>),

which rigorously calculates enrichment of this kind of split motifs provided that spacer length is conserved.

We very much appreciate the recommendation to use this analysis tool, which we were not previously aware of.

By performing the suggested *de novo* dyad motif discovery analysis on each of our ARF datasets we were able to independently verify enrichment for many of the spacing configurations that we observed in our initial directed analysis, confirming that certain clade A and clade B ARFs prefer different spacing/orientation configurations. The most frequently returned spacing pattern

identified by this program was the TGTC:GACA configuration with a spacing of 11 and 12 nucleotides and was found in both clade A and clade B ARFs. This configuration corresponds to the ER7,8 configuration that was captured in the crystal structure of Boer et al., 2014.

We have added a sentence in the main text to highlight the fact that *de novo* motif analysis uncovered many of the same motifs as our directed analysis. The text now reads:

“We also independently performed dyad motif discovery analysis (Defrance et al., 2008, Castro-Mondragon et al., 2016) was also performed and confirmed over-representation for many of these spacing patterns (Figure S5c).”

Methods

“*De novo* dyad discovery analysis was performed using RSAT plants^{37,38} with default parameters which allow the detection of spaced pairs of trinucleotides with up to 20 nucleotide spacing. Peaks containing only two TGTC instances were used as input. Representative motifs with identical orientations and spacing were collapsed for the logo representation shown in Figure S5c.”

5) I don't agree that Figure 3B proves that DNA binding affinity increases with the number of TGTC motifs in a peak. It rather shows that DAP is more efficient (captures more genomic fragments) if several TGTC sites are found.

We understand the reviewer's concern regarding this section. We have reworked the text already in response to reviewer #1's concern and have added an additional comment to the main text to acknowledge reviewer #2's concern.

“Taken together these data could suggest a model in which ARFs preferentially bind to motif clusters, resulting in higher affinity binding sites, or they may reflect the capture of more genomic fragments simply due to the higher density of TGTCs.”

6) The exact phases described in Figure 3D are hard to see. It would be easier if bars of 10bp are added on top of the heatmap, at least for clade A. For both clades it would be helpful to have conceptual models/plots of how dimers bind to those sites, if the *A. thalina* protein structures allow it.

We have added 10bp bars to the heatmap figure to highlight the 10bp phasing.

We would prefer to not include conceptual models because we feel it would be too speculative based on our data and the fact that a crystal structure is only available for the TGTC GACA (also referred to as the everted repeat orientation). Further experiments with mutations in key residues would be needed to support any speculative model.

7) Regarding Figure 4B, isn't 10 kb too wide a distance interval, particularly since gene bodies and proximal regions are enriched? What happens if 1kb is used instead? This is related to Fig4C. In either case, the barplot would improve if notches were added to define, for instance, 95% confidence intervals around the

medians.

We have selected a distance of 10kb to assess the number of peaks in auxin induced genes given that this is the distance at which the majority of ARF peaks, (particularly in the case of the cladeA ARFs). We performed similar calculations using 1kb, 2kb, and 5kb distances and always observe a significance difference in the number of peaks relative to random genes, however the differences are not as striking at these shorter distances. Hand curation of several previously studied genes such as the Aux/IAAs Bif1 and Bif4 (Galli et al., PNAS 2015) as well as many new examples added to the supplementary figures (FigS2C-K), demonstrates that peaks are often observed at distances greater than 5kb and we feel taking this into consideration is important.

The suggestion to add notches indicating 95% confidence intervals to the barplot greatly improves the figure panel and we appreciate the suggestion.

8) I have several comments about the machine learning section:

8.1) It's not clear the size of the training and validation sequences sets, nor whether training and validation sets were made by separate. In addition, the code should be published in a source code repo for the sake of transparency and reproducibility.

"The python code to implement this is available upon request" is not good enough because the code could change in the future with no logged changes.

We appreciate the reviewer's comments and agree wholeheartedly. We have uploaded all of our data, code, and trained models onto a github repository to ensure ease of access and reproducibility. The methods section now reads:

"The python code to implement this can be found at <https://github.com/arjunkhakhar/Maize-DapSeq-Machine-Learning>."

8.2) The authors say that "ARF16 binding peaks in the second DICE element differed slightly relative to those in the first DICE element, likely due to substantial sequence differences (Figure 6D)." Instead of this the authors should report objectively on the exact sequences differences at that locus.

An alignment showing the exact sequences the Mo17 second DICE element and B73 DICE element are shown in Fig. S7D, and peaks corresponding to each inbred line are indicated in orange. This is explained in the supplemental figure legend, however we neglected to reference it in the main text. We now reference this figure panel in the main text as follows:

"Interestingly, ARF16 binding peaks in the second DICE element differed slightly relative to those in the first DICE element, likely due to substantial sequence differences (Figure 6D, S7C)."

9) On page 4 the authors say that "Maize contains 32 expressed ARFs". In the current context of pangenomes and pantranscriptomes (<http://www.plantcell.org/content/early/2014/0131/tpc.113.119982>) the authors should say to which cultivar(s) this figure applies. is it B73?

This important point by the reviewer has been corrected and our text now indicates that we are referring to the first maize reference genome B73. "The maize B73 reference genome contains 32 expressed ARFs ^{19,20}."

REVIEWERS' COMMENTS:

Reviewer #1 (Remarks to the Author):

The authors have addressed adequately and thoroughly all my concerns. The manuscript has been strengthened significantly during the revision. It now provides a very convincing analysis of the binding landscape of ARFs. In addition it opens clear future directions for understanding further auxin signaling and for using this knowledge in crop selection.

Reviewer #2 (Remarks to the Author):

The authors have adequately addressed my previous concerns.

Reviewer #3 (Remarks to the Author):

The authors have done a very good job responding to my previous queries, thanks for that!

I still have a couple of minor suggestions:

Question 1) Is there a way to show that the GST fusion is not changing significantly DNA binding, or dimerization, at least for one ARF of each clade? Are there any ChIPseq datasets to compare?

The authors should explain in supplementary material and quote from the methods section their results of the HALO fusion tests.

Question 2) It would be helpful to see a distribution of length of affinity-captured genomic fragments, so that we know how hard is to discover DNA motifs within them.

It would be helpful to state in Methods that the mean genomic fragment size is ~200.

Response to Reviewer questions are shown as black font

REVIEWERS' COMMENTS:

Reviewer #1 (Remarks to the Author):

The authors have addressed adequately and thoroughly all my concerns. The manuscript has been strengthened significantly during the revision. It now provides a very convincing analysis of the binding landscape of ARFs. In addition it opens clear future directions for understanding further auxin signaling and for using this knowledge in crop selection.

Reviewer #2 (Remarks to the Author):

The authors have adequately addressed my previous concerns.

Reviewer #3 (Remarks to the Author):

The authors have done a very good job responding to my previous queries, thanks for that!

I still have a couple of minor suggestions:

Question 1) Is there a way to show that the GST fusion is not changing significantly DNA binding, or dimerization, at least for one ARF of each clade? Are there any ChIPseq datasets to compare?

The authors should explain in supplementary material and quote from the methods section their results of the HALO fusion tests.

1) To address reviewer #3 concerns and accommodate the specific request of the editor, we have added the following sentence to the methods section in order to account for the fact that we cannot explicitly state that the GST-tag has no effect.

“To rule out the possibility that artificial dimerization by the GST-tag could affect ARF DNA binding, we compared data from one clade A and one clade B GST-ARF fusion to *in vitro* expressed HALOtag-ARF-DBD fusions and found that over 93% of the HALO-tagged ARF peaks were present in the respective GST-tagged ARF datasets. While the HALO-ARF datasets had insufficient %RiP to be considered in our main analysis, we obtained similar results to GST-fusions in terms of motif enrichment, average number of TGTCs within peaks, and spacing patterns.”

Question 2) It would be helpful to see a distribution of length of affinity-captured genomic fragments, so that we know how hard is to discover DNA motifs within them.

It would be helpful to state in Methods that the mean genomic fragment size is ~200.

2) We apologize for our oversight in leaving out this important piece of information. We have added the following sentence to the methods section to state that the mean genomic fragment size is ~200bp.

“5µg of genomic DNA was diluted in EB (10mM Tris-HCl pH 8.5) and sonicated to 200bp fragments in a covaris S2 sonicator.”